



# When ancient numerical demons meet physics-informed machine learning: adjoint-based gradients for implicit differentiable modeling

Yalan Song[1], Wouter J. M. Knoben[2], Martyn P. Clark[2], Dapeng Feng[1,3], Kathryn Lawson[1], Chaopeng Shen[1]

[1]Civil and Environmental Engineering, The Pennsylvania State University, University Park, PA
[2]Department of Civil Engineering, Schulich School of Engineering, University of Calgary, Calgary, Alberta, Canada
[3]Department of Earth System Science, Stanford University, Stanford, CA

*Corresponding to*: Chaopeng Shen (cshen@engr.psu.edu), Yalan Song (yxs275@psu.edu)

**Abstract**. Recent advances in differentiable modeling, a genre of physics-informed machine learning that trains neural networks (NNs) together with process-based equations, has shown promise in enhancing hydrologic models' accuracy, interpretability, and knowledge-discovery potential. Current differentiable models are efficient for NN-based

parameter regionalization, but the simple explicit numerical schemes paired with sequential calculations (operator splitting) can incur large numerical errors whose impacts on models' representation power and learned parameters are not clear. Implicit schemes, however, cannot rely on automatic differentiation to calculate gradients due to potential issues of gradient vanishing and memory demand. Here we propose a "discretize-then-optimize" adjoint method to enable differentiable implicit numerical schemes for the first time for large-scale hydrologic modeling. The adjoint model demonstrates comprehensively improved performance, with Kling-Gupta efficiency coefficients, peak-flow

and low-flow metrics, and evapotranspiration that moderately surpass the already-competitive explicit model. Therefore, the previous sequential-calculation approach had a detrimental impact on the model's ability to represent hydrologic dynamics. Furthermore, with a structural update that describes capillary rise, the adjoint model can better describe baseflow in arid regions and also produce low and peak flows that outperform even pure machine learning methods such as long short-term memory networks. The adjoint model rectified some parameter distortions but did

not alter spatial parameter distributions, demonstrating the robustness of regionalized parameterization. Despite higher computational expenses and modest improvements, the adjoint model's success removes the barrier for complex implicit schemes to enrich differentiable modeling in hydrology.

## 1 Background

Accurate hydrologic predictions are crucial for effective water resource management around the world under a

changing climate (Hannah et al., 2011; Sivapalan et al., 2003). In recent years, deep learning models such as long short-term memory (LSTM) networks have gained traction in hydrology due to their high predictive performance in various applications, including streamflow prediction, soil moisture estimation, and the modeling of stream temperature and dissolved oxygen (Blöschl et al., 2019; Fang et al., 2017; Feng et al., 2020; Kratzert et al., 2019; Ouyang et al., 2021; Rahmani et al., 2021a, 2021b; Zhi et al., 2023). Despite their impressive capabilities, deep





learning models are often criticized for their limited interpretability and dependence on extensive observations. Additionally, they are unable to provide outputs for untrained variables (those not trained using observations as targets), e.g., evapotranspiration (ET), water storage, or snow water equivalent, which are of great interest to stakeholders but are not extensively observed.

In response to these limitations, alternative approaches that merge the process-based understanding of hydrologic systems with various genres of physics-informed machine learning techniques have been explored. These approaches include differentiable modeling (DM), interpretable machine learning approaches (Wang et al., 2021), post-processing (Frame et al., 2021), and embedding trained networks into existing models (Bennett and Nijssen, 2021). Notably, the neural networks must be pretrained using either observations or model simulations as the target. Not many approaches

allow interpretability, continued updating of the neural networks, and knowledge discovery at the same time. Recently, differentiable modeling (DM) (Shen et al., 2023) was proposed as a pathway to train neural networks (NNs) together with physical equations in an "end-to-end" fashion (Figure 1), where the NNs can provide parameters or unknown relationships for the process-based components (Tsai et al., 2021; Feng et al., 2023, 2022; Aboelyazeed et al., 2023; Bindas et al., 2023). When the model is "differentiable" (explained in the next paragraph), we have a "credit

assignment path" (Schmidhuber, 2015) between tunable parameters and the objective function, which enables efficient training of massive amounts of weights on big data based on outputs of the combined system. It also removes the need for direct supervising data for the output of the NN (although such data can be used as additional constraints when available), and enables the discovery of knowledge from data. To make the model differentiable, we prefer translating physical models onto differentiable platforms, which would guarantee the desired sensitivity as the physics are baked

into the model, over training an NN as a physical model surrogate. As the models are interpretable, they can be used to provide a full narrative of the physical processes and have the potential to discover scientific knowledge and unrecognized linkages from data. They also extrapolate better in space, especially in data-sparse regions, due to using process-based equations as the backbone of calculations and respecting assumed physical laws like conservation of mass (Feng et al., 2023).

Essentially, differentiable models aim to train coupled NNs (by optimizing their weights, w, in Figure 1) using gradient descent. In the end-to-end fashion, we must be able to calculate the gradients of model outputs with respect to NN weights along all the steps (physical equations and NN layers) in the model. Models supporting such gradient calculations are called "differentiable" models (Shen et al., 2023). Gradient descent is the only currently-known way to train NNs with massive amounts of weights on big data, and such computational infrastructure is very efficient,

especially when paired with parallel GPU (graphical processing units) processing and automatic differentiation (AD, explained further below). Modern machine learning platforms are built to support differentiability and NN training using AD, but there are other options as well. For example, while not the focus of DM, we can train NNs as surrogate models for physical models (minding all the complexities with maintaining the accuracy of a surrogate model as well as the fact that we cannot modify the internal functions of a surrogate model) and place it alongside a parameterization

network (Tsai et al., 2021). As another example, adjoint state methods have been developed to solve an accompanying equation to produce gradients of an equation-solving step (Chen et al., 2018).



While there have been significant advances in differentiable models, certain numerical approximations have been introduced to facilitate their easy implementation on machine learning platforms utilizing AD. These approximations can result in numerical errors, colloquially termed the "ancient numerical demon" (Clark and Kavetski, 2010; Kavetski and Clark, 2010) with implications for the calibrated parameter values. Such approximations encompass, but are not limited to, ad-hoc operator splitting (sequential operations), explicit numerical schemes without error control, and the use of threshold-like functions to avoid negative state variables, all of which can degrade the quality of gradient calculations for parameter optimization. Recently published differentiable hydrological models (Feng et al., 2023, 2022) mostly used these numerical approximations because of their straightforward implementation and a long legacy of usage (Aghakouchak and Habib, 2010; Beck et al., 2020; Bergström, 1992, 1976; Seibert and Vis, 2012). The numerical errors can be compensated for by pushing the calibrated parameters to a different part of the parameter space (Clark and Kavetski, 2010; Kavetski and Clark, 2010). With such compensation, the daily streamflow performance metrics such as Nash Sutcliffe model efficiency coefficient (NSE) can be kept high, but the interpretation of the results and the physical significance of the calibrated parameters are obscured. The implications of numerical errors were examined in detail before, where Kavetski and Clark (2010) addressed the issues using implicit schemes. However, the issue of explicit vs. implicit solvers has not been examined in the context of a regionalized parameterization scheme, especially a novel differentiable model relying on regionalized parameter learning with NNs, which applies implicit regional constraints. The extent to which numerical schemes can impact regionalized parameter distributions is unclear.

As an underpinning of deep learning, automatic differentiation (AD) decomposes complex calculations into a sequence of elementary arithmetic operations, and then applies the chain rule of differentiation to compute the derivative of the output with respect to its input variables. AD often needs to store some intermediate results or instructions, thus consuming memory. AD typically requires very little effort on the modeler's side besides writing the model in a forward mode (no need to provide gradient functions) on a machine learning platform like PyTorch, Tensorflow, JAX, or Julia, and is the obvious tool to calculate gradients for explicit models. However, if we want to improve a model's accuracy and stability using implicit solvers that require iterative steps, AD could run into the issue of having high overhead and excessive memory usage. Memory use is a significant issue for GPUs which are crucial to modern machine learning. Furthermore, tracking the gradients of the many iterative steps with AD may lead to the dreaded vanishing gradient problem (Hochreiter and Schmidhuber, 1997) facing recurrent neural network training, where the gradients become exceedingly small and prevent the NN weights from being effectively updated. These issues, unfortunately, make it challenging for differentiable models to employ implicit solvers that encompass a wealth of powerful and essential tools for solving some equations, e.g., those containing elliptic operators (Groundwater equations (Todd and Mays, 2004); Richards' equation (Richards, 1931); Saint-Venant equations (Strelkoff, 1970); heat equation (Bergman, 2011)) or systems of nonlinear equations (Aboelyazeed et al., 2023).

The adjoint method has been widely used for equation-constrained optimization (Cao et al., 2002) in various fields such as meteorology, oceanography, and geophysics, but only in recent years has it been applied for neural network training with differential equations (Rackauckas et al., 2021). Hydrologic modelers have also used the adjoint for data assimilation (Fisher and Andersson, 2001; White et al., 2003; Neupauer and Wilson, 2001; Liu and Gupta, 2007;





Castaings et al., 2009; Jay-Allemand et al., 2020; Bandai, 2022). Instead of automatically working through the elementary operations as AD does, the adjoint solves another accompanying equation (derived by the modeler based on the chain rule and the associative property of matrix multiplication) to rapidly produce the gradients of outputs with respect to inputs --- more precisely, it computes the vector-Jacobian product. However, the adjoint method has not yet been extensively explored in the context of large-scale, regionalized hydrologic simulations (White et al., 2003; Colleoni et al., 2022), which require mini-batch processing, high data throughput, and a long time for integration. It is unclear if the adjoint method is applicable in this scenario.

Adjoint methods can be defined at different levels. Simply put, the adjoint-state method is defined at the differential equation level (called "optimize-then-discretize"), involving the solution of a separate differential equation for the adjoint (Chen et al., 2018). However, we can also define it at lower functional levels, e.g., solving an adjoint equation for a specific operator inside a discretized numerical model (called "discretize-then-optimize") (Onken and Ruthotto, 2020). The latter is more naturally implemented along with the numerical algorithms to solve the forward problem.

In both machine learning and process-based modeling, an important point for consideration is whether the basic architecture has the expressive power to represent the phenomenon of interest. Deep networks can approximate extremely complex functions, due to the enormous amount of weights that can be trained, along with their generic architecture (Hornik et al., 1989). For process-based models (or hybrid differentiable models), structural deficiencies may lead to problematic behaviors that cannot be remediated (even with the help of highly flexible NNs for parameterization). It is unclear whether the difference between implicit and explicit solvers can lead to differences in representation, and whether differentiable models can help us identify structural deficiencies.

In this work, we proposed the application of the "discretize-then-optimize" adjoint method to implement implicit numerical schemes in differentiable hydrologic models (referred to as implicit adjoint-based models or "adjoint models" for brevity, also denoted as δHBV.adj.). We then compared them to the existing differentiable models with explicit Euler time stepping and sequential operations (referred to as explicit sequential models or "sequential models" for brevity and are denoted as δHBV). We investigated the impacts of these methods on hydrologic model performance and parameter distributions. Furthermore, we examined the potential for these adjoint-based methods to enhance the performance of differentiable models, bringing them closer to or surpassing the performance of state-of-the-art LSTM models. We sought answers for the following questions:

1.  *Can we support implicit numerical solvers in large-scale differentiable hydrologic modeling, and what are their implications for performance and computational efficiency?*

2.  *Do implicit and sequential models have different representation power, i.e., does the sequential-calculation approach result in errors that prevent it from accurately representing certain aspects of hydrologic dynamics?*

3.  *Do we get very different parameter distributions with the implicit (adjoint) model than with the sequential model at the regional and local scales?*

The full version name of the adjoint model is δHBV.adj-CAMELS-hydroDL where "δ" indicates differentiable modeling, "adj" represents adjoint, "CAMELS" represents the training dataset, and "hydroDL" stands for software implementation.





## 2 Data and Methods

As a high-level summary, the differentiable model couples an LSTM network to a conceptual hydrologic model, Hydrologiska Byråns Vattenbalansavdelning (HBV), and trains them together in an end-to-end fashion (from the input of LSTM to the output of the HBV) on daily discharge data on 671 basins in the conterminous United States (CONUS). The LSTM, which provides static or dynamic parameters for HBV, is not trained directly with pre-calibrated parameters but is jointly trained with HBV using the discharge observations. This joint training, where the LSTM weights are updated, is supported by either AD for the explicit HBV model or the adjoint method for the implicit HBV model. The conceptual frameworks of the two models are similar, and the differences between AD and the adjoint only pertains to how the gradients are obtained during backpropagation through the solving of HBV's equations. However, due to gradient vanishing and memory issues arising from the required numerical iterations, the implicit model simply cannot be supported by AD for backpropagation. In addition to the implementation of the implicit scheme, this work also evaluated a structural change (incorporating capillary rise) to HBV, based on insights acquired during joint training. We compared the explicit (sequential), implicit, improved explicit (sequential) and improved implicit differentiable models and a direct LSTM simulation of streamflow in terms of various streamflow metrics, and the differentiable models are also benchmarked in terms of ET and baseflow fraction simulations. In the following, we describe the different parts of the framework as well as discuss the differences between AD and the adjoint method in more detail.

### 2.1 Datasets

We utilized the Catchment Attributes and Meteorology for Large-sample Studies (CAMELS) data set (Addor et al., 2017; Newman et al., 2014) for this study. This dataset comprises basin-averaged hydrometeorological time series, catchment attributes, and streamflow observations from the United States Geological Survey (USGS) for 671 catchments across the CONUS. The majority of its daily streamflow observations span from 1980 to 2014. For our study, the meteorological forcing data was sourced from Daily Surface Weather Data on a 1-km Grid for North America (Daymet) Version 4 (Thornton et al., 2020). From the CAMELS dataset, we incorporated catchment attributes such as topography, climate patterns, land cover, soil, and geological characteristics as inputs to our models (Table A1).

We compared our model simulation with the streamflow observation, as well as the streamflow simulation from a traditional process-based model, SAC-SMA (Sacramento Soil Moisture Accounting), with the model calibrated by the National Weather Service. The simulation results of SAC-SMA are provided in CAMELS. To assess the accuracy of predicted intermediate variables, we also employed the Baseflow Index (BFI) from the CAMELS dataset and evapotranspiration (ET) data from the MOD16A2 dataset (Running et al., 2017). BFI is obtained by applying Lyne and Hollick filters with warm-up periods to streamflow hydrographs (Ladson et al., 2013). The MOD16A2 Version 6 Evapotranspiration/Latent Heat Flux product provides 8-day composites at a 500-meter pixel resolution and is aggregated to the average values at the basin levels. The algorithm used for the MOD16A2 dataset relies on the Penman-Monteith equation logic, incorporating daily meteorological reanalysis data and Moderate Resolution



Imaging Spectroradiometer (MODIS) remote sensing inputs like vegetation dynamics, albedo, and land cover (Monteith, 1965; Mu et al., 2011; Running et al., 2017).

## 2.2. Models

### 2.2.1 Differentiable, Learnable, Regionalized Process-Based Model

This study utilized the differentiable regionalized process-based model framework presented in Feng et al. (2022), which employs the HBV model as its backbone and utilizes LSTM for parameter regionalization. The motivation for making the model differentiable is so that it can train connected NNs in an "end-to-end" manner to learn robust and complex relationships from big data, which can provide indirect supervision to the NNs. As stated above, the model was already made programmatically "differentiable" because it was implemented on the PyTorch ML platform, so gradient information can backpropagate through it. However, it previously was only integrated in time using the explicit Euler method with no error control. The HBV model is a conceptual model that uses a set of linked storage components representing processes like snow accumulation and melt, soil moisture dynamics, and river routing to forecast river discharge. To summarize the NN-HBV coupling succinctly, the differentiable model based on HBV can be written as:

$$\theta = g_w(x, A) \tag{1}$$

where $\theta$ represents HBV physical parameters, $A$ contains 35 static attributes such as topography, climate, soil texture, land cover, and geology (Table A1 in Appendix), x represents the meteorological forcings and $g_w$ is a parameterization neural network that seeks to capture the prevalent relationship between the input data and the HBV parameters ($w$ represents the weights of the neural network). $\theta$ can be formulated as being either static-in-time or time-dependent, where new values are obtained for every day of the simulation. More details about the data can be found in Feng et al. (2022). The HBV forward simulation is succinctly written as

$$Q = HBV(x, \theta) \tag{2}$$

where $Q$ is the simulated streamflow. The HBV model utilizes three primary forcing variables: precipitation ($P$), temperature ($T$), and potential evapotranspiration ($E_P$). The Hargreaves (1994) method, which considers mean, maximum, and minimum temperatures along with latitudes, is employed to estimate $E_P$, representing the total evaporative demand. The same forcings, $X = \{P, T, E_P\}$, was used in $g_w$. It should be noted that HBV only serves as an example, and other hydrologic models (Knoben et al., 2019) can be similarly employed.

### 2.2.2 Hydrologiska Byråns Vattenbalansavdelning model

The HBV model employs a framework that includes five water storages and associated fluxes to encapsulate the primary hydrological processes within a catchment. It can simulate hydrologic variables, including soil moisture, groundwater storage, evapotranspiration, quick flow, baseflow, and streamflow. It consists of four main modules to classify all storages and fluxes as shown in Figure. 1:





Snow Accumulation and Melt: This module uses a temperature-index method to distinguish between rainfall and snowfall and to simulate the snow accumulation and melt processes.

$$\frac{dS_p}{dt} = P_s + R_{fz} - S_{smelt} \tag{3}$$

$$P_s = P \ \text{if} \ T < \theta_{TT}, \text{otherwise} \ 0 \tag{4}$$

$$R_{fz} = (\theta_{TT} - T)\theta_{DD}\theta_{rfz} \tag{5}$$

$$s_{melt} = (T - \theta_{TT}) \ \theta_{DD} \tag{6}$$

where $t$ is time; $S_p$ is the current snow storage [mm]; $P_s$ is the precipitation as snow [mm/day]; $R_{fz}$ is the refreezing of liquid snow [mm/day]; $s_{melt}$ is the snowmelt as water equivalent [mm/day]; $\theta_{TT}$, $\theta_{DD}$, and $\theta_{rfz}$ are threshold temperature for snowfall [°C], degree-day factor [mm°C-1day-1], and refreezing coefficient [-], respectively.

$$\frac{dS_{liq}}{dt} = s_{melt} - R_{fz} - I_{snow} \tag{7}$$

$$I_{snow} = s_{liq} - \theta_{CWH}S_p \tag{8}$$

where $I_{snow}$ is the snowmelt infiltration to soil moisture [mm/day]; $S_{liq}$ is the liquid water content in the snowpack [mm]; and $\theta_{CWH}$ is the water holding capacity as a fraction of the current snowpack [-].

Soil Moisture and Evapotranspiration: The model features a simple soil moisture accounting scheme where precipitation and snowmelt infiltration can either contribute to evapotranspiration or runoff. Potential evapotranspiration, typically calculated externally (e.g., using the Hargreaves method (Hargreaves, 1994)), limits the actual evapotranspiration from the soil storage.

$$\frac{dS_S}{dt} = I_{snow} + P_r - P_{eff} - E_x - E_T(+C_r) \tag{9}$$

$$P_r = P \ \text{if} \ T > \theta_{TT} \ \text{otherwise} \ 0 \tag{10}$$

$$P_{eff} = \min\left(\left(\frac{S_S}{\theta_{FC}}\right)^\beta, 1\right)(P_r + I_{snow}) \tag{11}$$

$$E_x = (S_S - \theta_{FC})/dt \tag{12}$$

$$E_T = \min\left(\left(\frac{S_S}{\theta_{FC}\theta_{LP}}\right)^\gamma, 1\right)E_P \tag{13}$$

$$C_r = \theta_C * S_{LZ} * \left(1 - \frac{S_S}{\theta_{NDC} * \theta_{FC}}\right) \tag{14}$$





where $S_S$ is the current storage in soil moisture [mm]; $S_{LZ}$ is the current storage in the lower subsurface zone [mm/day]; $P_r$ is the precipitation as rain [mm/day]; $P_{eff}$ is the effective flow to the upper subsurface zone [mm/day]; $E_x$ is the rainfall excess [mm/day]; $E_T$ is the actual evapotranspiration [mm/day]; $C_r$ is the capillary rise from the lower subsurface zone; $\theta_c$ is a time parameter [day-1]; $\theta_{FC}$ is the maximum soil moisture (field capacity) [mm]; $\theta_{NDC}$ is the fraction ratio of the field capacity [-] and is set to a constant value of 1 in the discrete model to prevent gradient explosion, while in the adjoint model, it is treated as a static parameter to be learned; $\theta_{LP}$ is the vegetation wilting point [-]; $\beta$ is a parameter influencing the shape of the soil moisture function [-]; $\gamma$ is a parameter influencing the shape of the evapotranspiration function [-].

Runoff Generation: Runoff in the HBV model is represented by three components - two quick flows (near surface flow and interflow), and delayed runoff (or baseflow).

$$\frac{dS_{UZ}}{dt} = P_{eff} + E_x - perc - Q_0 - Q_1 \tag{15}$$

$$Perc = \min(\theta_{perc}, S_{UZ}/dt) \tag{16}$$

$$Q_0 = \theta_{K_0}(S_{UZ} - \theta_{UZL}) \tag{17}$$

$$Q_1 = \theta_{K_1}S_{UZ} \tag{18}$$

$$\frac{dS_{LZ}}{dt} = Perc - Q_2(-C_r) \tag{19}$$

$$Q_2 = \theta_{K_0}S_{LZ} \tag{20}$$

where $S_{UZ}$ is the current storage in the upper subsurface zone [mm]; $perc$ is the percolation to the lower subsurface zone [mm/day]; $Q_0$, $Q_1$, and $Q_2$ are the near surface flow [mm/day], interflow [mm/day], and baseflow [mm/day], respectively; $\theta_{perc}$ is the percolation flow rate [mm/day]; $\theta_{K_0}$, $\theta_{K_1}$, and $\theta_{K_2}$ are the recession coefficients [day-1].

Basin-scale routing: We employ a gamma function to simulate the flow routing through rivers and lakes within the catchment, leading to the simulated discharge at the catchment outlet.

$$Q(t) = \int_0^t \xi(s)Q'(t-s)ds \tag{21}$$

$$\xi(s) = \frac{1}{\Gamma(\theta_a)\theta_b{}^{\theta_a}}s^{\theta_a-1}e^{-\frac{1}{\theta_b}} \tag{22}$$

where $Q' = Q_0 + Q_1 + Q_2$; $Q$ is the simulated streamflow at the catchment outlet; $\theta_a$ [-] and $\theta_b$[-] are two routing parameters.

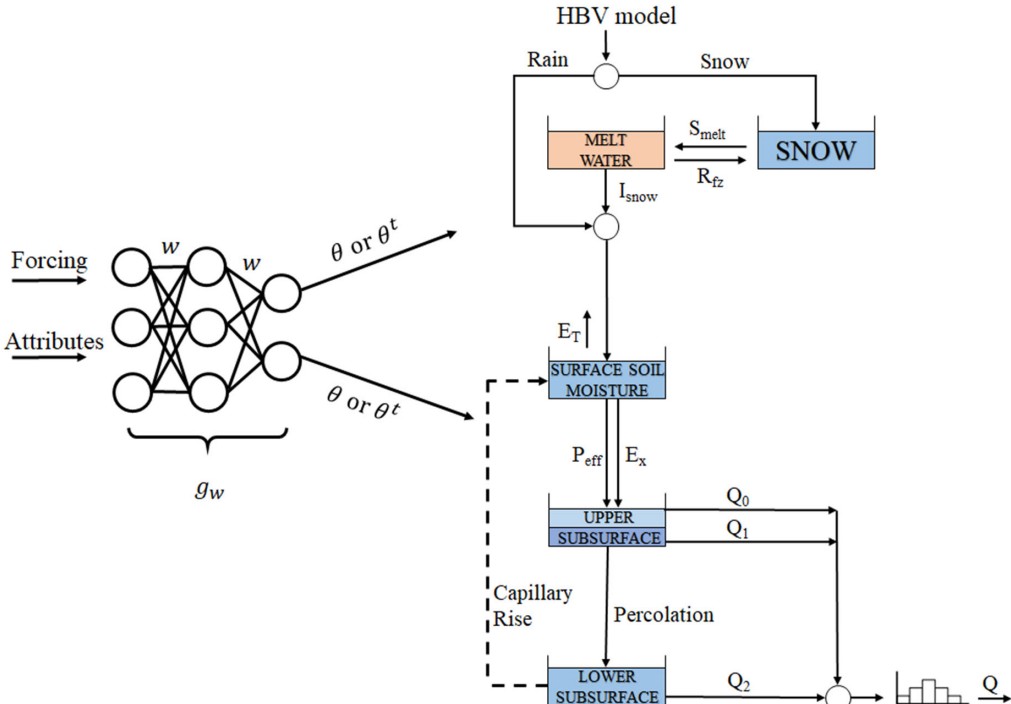

**245**

**Figure 1:** A schematic view of the HBV model. $g_w$ is a parameterization neural network that seeks to capture the prevalent relationship between raw input data and the HBV parameters (w represents the weights of the neural network). $\theta$ and $\theta^t$ are the static and dynamic HBV parameters, respectively. $I_{snow}$ is the snowmelt infiltration to soil moisture, $R_{fz}$ is the refreezing of liquid snow, and $s_{melt}$ is the snowmelt as water equivalent. $E_T$ is the actual evapotranspiration, $P_{eff}$ is the

**250** effective flow to the upper subsurface zone, and $E_x$ is the rainfall excess. $Q_0$, $Q_1$, and $Q_2$ are the near surface flow, interflow, and baseflow, respectively. Q is the simulated streamflow at the catchment outlet.

In this work, we investigated two distinct HBV structures. The first structure replicates the structure employed in Feng et al. (2022) adapted from the HBV structure used in Beck et al. (2020). A primary limitation of this structure is its

**255** inability to represent the depletion of storages and the occurrence of zero flow. In scenarios where precipitation events are minimal and soil moisture is not entirely depleted, there always exists a recharge flow directed into the groundwater compartments, consequently producing a baseflow. To alleviate this limitation, according to previous experiences (Knoben et al., 2019), various strategies can be employed: (1) applying a threshold-based function to Q1 and Q2, (2) constraining the effective rainfall and excess volume, (3) adapting the ET functions, (4) introducing a sink flux in the

**260** lower subsurface zone to directly remove water, and (5) incorporating a capillary flux to redistribute water among soil zones. Among these strategies, we chose to incorporate a capillary flux from the lower subsurface zone to the surface soil, as illustrated by the dashed arrow in Figure 1, in order to avoid introducing threshold-like functions, which would complicate gradient calculations. Notably, this is not the typical capillary flux from the upper subsurface zone. Similarly, the reciprocal flux between different soil zones is profoundly influenced by the operation order, making it

**265** more susceptible to numerical errors in the sequential models. The capillary flux can increase the evapotranspiration, especially when surface soil moisture is at a diminished level, thereby moderating the baseflow. This flux (Equation



14) is mathematically accounted for by additional terms in brackets of Equations 9 and 20, whereby it is subtracted from the lower subsurface soil zone and added to the upper soil zone. The sequential and adjoint model incorporating the capillary flux are referred to as "the sequential improved model ($\delta$HBV improved model)" and "the adjoint

improved model ($\delta$HBV.adj improved model)", respectively.

### 2.2.3 Long short-term memory network

Long Short-Term Memory (LSTM) is a recurrent neural network (RNN) designed for identifying patterns in long time series. While traditional RNNs face challenges like vanishing or exploding gradients (Hochreiter and Schmidhuber, 1997), LSTM alleviates these issues using a distinct cell architecture with input, forget, and output gates. These gates

modulate information flow, rendering LSTM especially effective for tasks like time series forecasting and sequence-to-sequence modeling. Given LSTM's proficiency in capturing temporal dynamics, it serves two primary functions in this study: direct streamflow prediction, and regionalized parameterization.

Direct streamflow prediction using LSTM:

$$Q = LSTM(x, A, w) \tag{23}$$

This model serves as a benchmark. The meteorological forcings, $x$, used in the pure LSTM includes precipitation,

solar radiation, max and min temperature, and vapor pressure. Attributes, $A$, includes topography, climate, soil texture, land cover, and geology. $w$ are the LSTM weights to be trained for streamflow prediction. More details about the pure LSTM streamflow model can be found in Kratzert et al. (2019) and it is referred to here as the LSTM model. We used the code from Kratzert et al. (2019) and ran the model in the same training and testing periods as the HBV models. Although we have a separate implementation that generated similar performance (Feng et al., 2020; 2021), here we

include just one LSTM model from a third party for cross comparability. While LSTM offers exceptional accuracy in streamflow prediction, its application in hydrologic modeling presents interpretability challenges and it does not produce intermediate physical states or fluxes.

Regionalized parameterization:

LSTM can also serve as a parameter learning function, referred to as gw in the DM framework:

$$\theta \ or \ \theta^t = LSTM(x, A, w) \tag{24}$$

where the parameters learned can be either static ($\theta$) or time-dynamic ($\theta^t$). The forcings, $x$, only include precipitation, temperature, and potential evapotranspiration used in HBV (same as Feng et al. (2022)). A includes the same 35 static attributes used in the pure LSTM model, listed in Table A1. $w$ represents the LSTM weights to be trained for HBV parameter estimation.

When employed for regionalized parameterization (in this context, this means all available training sites are employed

to train one network), LSTM establishes a correlation between input data and HBV parameters. By learning from a dataset with 671 basins, it tries to learn the implications of basin characteristics, making it applicable to ungauged basins. The HBV parameters used in Eq. 3 - Eq. 22 can be treated as static ($\theta$) or dynamic parameters ($\theta^t$). When treated as static, the same parameter value is used throughout the HBV simulation, whereas dynamic parameterization





(DP) provides a time series of parameters that differ for each basin and each day. In Feng et al. (2022), the DP approach

was adopted for parameter $\gamma$, which was intended to reflect the impacts of vegetation on ET. Additionally, the shape coefficient, $\beta$, was set to be dynamic to account for the nonlinear relationship between surface soil moisture and effective rainfall. In the adjoint model, besides $\gamma$ and $\beta$, the field capacity, $\theta_{FC}$ is also treated as dynamic to enhance the adaptability of the model.

### 2.2.4 Backpropagation with a coupled NN and process-based model

Within the framework of the differentiable model, the HBV model's parameterization is achieved by optimizing the weights, $w_i$, of the LSTM. This process learns the relationship between the input big data and the optimal parameterization using gradient descent:

$$w^{n+1} = w^n - \alpha \frac{dL}{dw^n} \tag{25}$$

where $\alpha$ is the learning rate (a model hyperparameter) and can be updated automatically by the optimizer, AdaDelta (Zeiler, 2012); $L$ is the loss function that evaluates the discrepancy between the simulated and observed streamflow

based on root-mean-square error (RMSE), conducted on a mini-batch of basins during the training process; n is the iteration number.

The gradient $\frac{dL}{dw}$ can be decomposed into multiple terms using the chain rule:

$$\frac{dL}{dw} = \frac{\partial L}{\partial w} + \frac{\partial L}{\partial \theta}\frac{d\theta}{dw} = \frac{\partial L}{\partial w} + \left(\frac{\partial L}{\partial Q}\frac{\partial Q}{\partial S}\frac{dS}{d\theta}\right)\frac{d\theta}{dw} \tag{26}$$

where $\frac{\partial L}{\partial w}$ represents the gradient of regularization terms applied to the weights in the loss function and $\frac{\partial L}{\partial \theta}$ represents the gradient of the loss function with respect to the HBV parameters ($\theta$), which encompasses the backpropagation

steps through the loss function associated with the streamflow ($\frac{\partial L}{\partial Q}$) and HBV functions ($\frac{\partial Q}{\partial S}\frac{dS}{d\theta}$). $\frac{d\theta}{dw}$ represents the gradient of the HBV parameters with respect to the LSTM weights. During backpropagation, we automatically obtain the gradient vector $\frac{dL}{dw}$ as the program tracks through each function and resolves the gradients from left to right of the chain rule terms in Equation 26.

### 2.2.5 Adjoint-based implicit scheme

The HBV model, like many hydrological models, relies on a set of ordinary differential equations (ODEs) to simulate processes. Following many previous hydrologic modeling studies, Feng et al. (2022) solved the HBV model by describing each process in a sequential manner with a daily time step (sequential model). They managed the fluxes by sequentially adding to or subtracting from the water storages, limiting the depletion or saturation of storages with threshold functions, and ensuring that the storages were updated following each individual process. It remains unclear

how the explicit and sequential approach to solving ODEs influences the parameter estimation and internal fluxes, subsequently altering the prediction of streamflow.





To enhance numerical accuracy, this study utilized an adjoint-based implicit numerical scheme to solve the ODEs simultaneously and implicitly. The gradient components, $\frac{\partial L}{\partial Q}, \frac{\partial Q}{\partial S}$, and $\frac{d\theta}{dw}$, in Eq. 26 can be easily handled by AD. The

other gradient component, $\frac{dS}{d\theta}$, is challenging for an implicit solver due to the numerous iterations and matrix solving required (Eq. 2).

Consistent with the approach adopted in MARRMoT (Knoben et al., 2019), in one time step, the time derivatives of all ODEs are discretized using a first-order backward Euler implicit scheme:

$$\frac{S^t - S^{t-1}}{\Delta t} = f(S^t, \theta^t) \tag{27}$$

This can be reformulated as a nonlinear equation:

$$F(S^t, \theta^t) = f(S^t, \theta^t) - \frac{S^t - S^{t-1}}{\Delta t} = 0 \tag{28}$$

The Newton-Raphson method is used to solve Eq. (27):

$$S^{t,n} = S^{t,n-1} - \frac{F(S^{t,n-1}, \theta^{t,n-1})}{F'_S(S^{t,n-1}, \theta^{t,n-1})} \tag{29}$$

where $n$ is the iteration number and $S$ simply refers to a generic storage. Typically, Equation 29 is computed for many iterations until convergence, but the number of iterations poses a challenge to AD and memory as discussed above. To avoid applying AD throughout the iterations, we differentiate $F(S, \theta) = 0$ with respect to $\theta$:

$$\frac{\partial F}{\partial \theta} + \frac{\partial F}{\partial S}\frac{dS}{d\theta} = 0 \tag{30}$$

Thus, we obtain the following equation.

$$\frac{dS}{d\theta} = -(\frac{\partial F}{\partial S})^{-1}\frac{\partial F}{\partial \theta} \tag{31}$$

Substituting Eq. 31 into Eq. 26, the gradient of weights of LSTM becomes:

$$\frac{dL}{dw} = \frac{\partial L}{\partial w} + \frac{dL}{d\theta}\frac{d\theta}{dw} = \frac{\partial L}{\partial w} - (\frac{\partial L}{\partial S}(\frac{\partial F}{\partial S})^{-1}\frac{\partial F}{\partial \theta})\frac{d\theta}{dw} \tag{32}$$

We seek to solve for the "adjoint", $\lambda$, which satisfies:

$$(\frac{\partial F}{\partial S})^T\lambda = -(\frac{\partial L}{\partial S})^T \tag{33}$$

Here the adjoint is a so-called "vector-Jacobian product", where $(\frac{\partial L}{\partial S})^T$ is the vector and $((\frac{\partial F}{\partial S})^T)^{-1}$ is the Jacobian matrix.

Upon obtaining the adjoint $\lambda$, we substitute it into Eq. 32 :

$$\frac{dL}{dw} = \frac{\partial L}{\partial w} + \frac{dL}{d\theta}\frac{d\theta}{dw} = \frac{\partial L}{\partial w} + (\lambda^T\frac{\partial F}{\partial \theta})\frac{d\theta}{dw} \tag{34}$$





While solving for the adjoint requires solving a matrix, the adjoint method bypasses the need for direct AD through all Newton iterations. Only after the Newton iteration converges and the solution is obtained do we need to compute the vector and the Jacobian and solve the Jacobian matrix, thus greatly reducing the amount of information that requires AD. Furthermore, in our implementation, the adjoints for all the basins in the minibatch are computed in parallel to permit rapid training on the GPU.

**2.2.6 Metrics, Model Training, and Hyperparameters**

The training phase employed data spanning 15 years, from 1 October 1980 to 30 September 1995, while the performance evaluation was conducted on data spanning another 15 years, from 1 October 1995 to 30 September 2010. In all cases, one neural network was trained on all the training basins with all training data. The hyperparameters of the LSTM unit were inherited from Feng et al. (2022). A hidden state of 256, a mini-batch size of 100, and a time

series length of 365 days were used to train the models. The model was trained to minimize an objective function (loss function) based on root-mean-square error (RMSE) across all basins in a mini-batch:

$$Loss = (1.0 - \alpha_l)\sqrt{\frac{\sum_{b=1}^{B}\sum_{t=1}^{T}(Q - Q*)^2}{B*T}} + \alpha_l\sqrt{\frac{\sum_{b=1}^{B}\sum_{t=1}^{T}(\hat{Q} - \widehat{Q*})^2}{B*T}} \tag{35}$$

$$\hat{Q} = log_{10}(\sqrt{Q + \epsilon} + 0.1) \tag{36}$$

where $B$ is the number of basins (mini-batch size), $T$ is the number of days involved in the training (time series length), and $\hat{Q}$ is the log-transformed streamflow (transformation done to better represent the low flows in the training data). $\epsilon$ is a small value $(1*10^{-6})$ to stabilize the gradient calculation. $\alpha_l$ is a weight parameter to balance the model's

performance between high flow and low flow, where a large value of $\alpha_l$ intends to improve the low flow performance. Here we set $\alpha_l$ to 0.25, which was manually tuned in Feng et al. (2022).

To evaluate model performance, the Nash-Sutcliffe model efficiency coefficient (NSE; Nash & Sutcliffe (1970)), the Kling-Gupta model efficiency coefficient (KGE; Gupta et al., (2009)), and the low flow and peak flow related hydrological signatures were computed for streamflow as well as metrics for other hydrological variables such as ET

and baseflow. The metrics used to evaluate the all model performance were:

- NSE: The NSE metric was derived from the ratio of the error variance of the modeled time series to the variance of the observed time series, with a value of 1 indicating a perfect model and 0 indicating performance equivalent to using the long-term mean value as the prediction.

- KGE: The KGE metric considers correlation, bias, and flow variability error, with a perfect simulation having

a value of 1.

- Low flow RMSE: The low flow RMSE represents the RMSE of the bottom 30% of the streamflow range.

- Peak flow RMSE: The peak flow RMSE represents the RMSE of the top 2% of the streamflow range.

- Absolute FLV: The percent of absolute bias of the bottom 30% ("low") flow range. That is, the sum of the absolute bias of the low flow divided by the sum of the low flow values.





•    Absolute FHV: The percent of absolute bias of the top 2% ("peak") flow range. That is, the sum of the absolute bias of peak flow divided by the sum of peak flow values.

•    Baseflow index spatial correlation: The correlation between simulated BFI ($Q_2/Q$) and BFI from the CAMELS derived from Ladson et al. (2013) across all basins in a spatial context.

•    Temporal ET simulation NSE: NSE of the ET time series from the models compared against ET data from
the MODIS satellite mission.

### 3. Results and Discussion

In this section, we first examine the overall performance of the adjoint model in comparison with the sequential model and direct LSTM simulation. Then, we examine the impact of a structural change (adding capillary rise to improve baseflow performance) on the sequential and adjoint models. Finally, we examine how using an explicit sequential
solution or implicit solutions impacts the spatial distribution of parameters produced by the regionalized parameterization network.

### 3.1. Adjoint model

Before making any structural changes, the adjoint model already demonstrated a highly competitive streamflow prediction performance overall --- its KGE, high-flow, and low-flow metrics are all modestly better than those of the
sequential model (Table 1). For KGE, the adjoint model (0.75) is higher than the sequential model (0.73) but lower than LSTM (0.77). In terms of peak-flow RMSE (lower is better), the adjoint model scored 2.47 mm/day, lower (and thus better) than the 2.56 mm/day scored by both the sequential model and LSTM, which was in turn noticeably lower than SAC-SMA's 3.19 mm/day. In terms of low-flow RMSE, the adjoint model's performance (0.048 mm/day) was lower than that of the sequential model (0.074 mm/day) and LSTM (0.055 mm/day).
Surprisingly, the adjoint model ($\delta$HBV.adj) even slightly surpassed the LSTM in peak-flow accuracy, reducing the median high-flow RMSE by 0.1 mm/day and the median FHV by 0.2%. While this difference is small, we remind the readers that the metrics are extremely difficult to improve at this level and we have not noticed better high-flow metrics elsewhere for this benchmark. The advance may be attributable to $\delta$HBV.adj's mass balance preservation which forces the model to more accurately represent extreme values. It's worthwhile to mention that while $\delta$HBV.adj is competitive
with LSTM at the median on CAMELS basins, as illustrated earlier, it was outperformed by LSTM for the low-KGE basins (lower part  in Figure 2 where NSE or KGE is in [0,0.5]). One possibility is that rainfall data have significant predictable bias and errors for these basins (see more discussion in Section 4). On a side note, this comparison also highlights that a single metric like KGE or NSE may not tell the full story.

The adjoint model improved predictions for other hydrological variables not employed as training targets (and thus
cannot be directly simulated by LSTM), including baseflow and ET. The spatial correlation of the simulated Base Flow Index (BFI) ($Q_2/Q$) was enhanced to 0.83 in comparison with the sequential model's correlation of 0.76 (Table 2). This improvement is consistent with the adjoint model's superior ability to capture low flows. The correlation of simulated ET to the MODIS product was increased from 0.59 with the sequential model to 0.61 with the implicit



(adjoint) model. Both BFI and MODIS products are only alternative estimates, but they are derived using different
methods and MODIS utilizes independent information, and thus a better agreement is nonetheless an indication of
better model behavior.

As the implicit model comprehensively improved the streamflow simulation (high and low flows, uncalibrated
variables), we conclude that the numerical errors of the sequential model, introduced by the dependence on the order
of calculations, have a noticeable negative impact on the model's ability to represent hydrologic dynamics and fit
observations. The differences are admittedly small, but one should not expect major gaps here because the sequential
model was already highly competitive and did not leave too much room for improvement (Feng et al., 2022; 2023). It
is also worthwhile mentioning that the small differences in the median metrics could manifest as large differences in
capturing some peak events (Figure 5).

Probing further into the low-flow issue, the sequential model seemed to have significant structural deficiencies in
representing low flows, which were remediated by using the implicit solver. The sequential model's FLV values were
much larger than LSTM's (Figures 3a & 4c). With the implicit solver, the adjoint model reduced FLV for a number
of regions: (i) on the Great Plains; (ii) in Indiana/Ohio (south of Lake Michigan); and (iii) some basins in the southeast
including Florida (Figure 3e). Therefore, the numerical errors with the sequential model exerted a substantial negative
impact on the model's ability to accurately represent baseflow. The adjoint model mitigated the overestimation of zero
and near-zero flows in arid areas (as seen in Figure 5, site i) and also corrected the underestimation of the recession
limb (refer to Figure 5, site ii).

We suspect the above-highlighted regions are where effective flow strongly competes with runoff and ET, and thus
the order of calculations has large impacts on the separation of fluxes. On the Great Plains, the precipitation tended to
be in sync with potential ET, and both were high in summer months (Fang and Shen, 2017) --- when the sequential
HBV calculates ET before effective flow, there can be significantly less effective flow than if effective flow was
computed first. The belt of Indiana/Ohio has compacted soil with high bulk density and difficulty with drainage and
thus a shallow water table, which also exists for Florida due to the low relief. In all of these cases, there is competition
between effective flow and other processes (ET or excess rainfall): calculating excess first could generate more excess
volume than would be if ET is calculated first. In the arid southwest, the competition between effective flow and ET
is also important. The implicit scheme mitigates this problem by solving two operators simultaneously while avoiding
overshooting fluxes or stability issues, which enables a better fit to the data.

**Table 1: Summary of statistical streamflow metrics for all models in the testing period using Daymet meteorological forcing data. We used the code from Kratzert et al. (2019) for cross-group comparability.**

| Model | Median NSE | Median KGE | Median absolute (non-absolute) FLV (%) | Median absolute (non-absolute) FHV (%) | Median low flow RMSE (mm/day) | Median peak flow RMSE (mm/day) | Dynamic parameters |
|---|---|---|---|---|---|---|---|
| LSTM | **0.73** | **0.77** | 40.59 (29.70) | 13.46 (-4.19) | 0.055 | 2.56 | |





| SAC-SMA | 0.66 | 0.73 | 59.40 (46.96) | 17.55 (-9.79) | 0.081 | 3.19 | - |
| $\delta$HBV | **0.73** | 0.73 | 56.53 (50.93) | 15.29 (-8.89) | 0.074 | 2.56 | $\gamma, \beta$ |
| $\delta$HBV.adj | 0.72 | 0.75 | 43.29 (37.61) | **13.25 (-4.33)** | 0.048 | **2.47** | $\gamma, \beta, \theta_{FC}$ |
| $\delta$HBV improved | **0.73** | 0.75 | **35.69 (21.09)** | 15.45 (-10.61) | 0.049 | 2.72 | $\gamma, \beta$ |
| $\delta$HBV.adj improved | **0.73** | 0.76 | 37.63 (28.63) | 14.36 (-6.04) | **0.047** | 2.59 | $\gamma, \beta, \theta_{FC}$ |


**Table 2: Summary of the statistical hydrological signatures of all models in the testing period**

| Methods | Baseflow index spatial correlation | Median NSE of temporal ET simulation |
|---|---|---|
| LSTM | - | - |
| SAC-SMA | - | - |
| $\delta$HBV | 0.76 | 0.59 |
| $\delta$HBV.adj | 0.83 | **0.61** |
| $\delta$HBV improved | 0.80 | 0.54 |
| $\delta$HBV.adj improved | **0.86** | 0.6 |

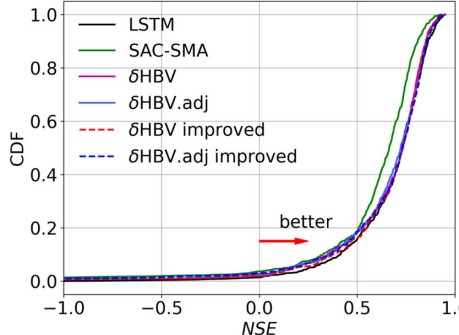
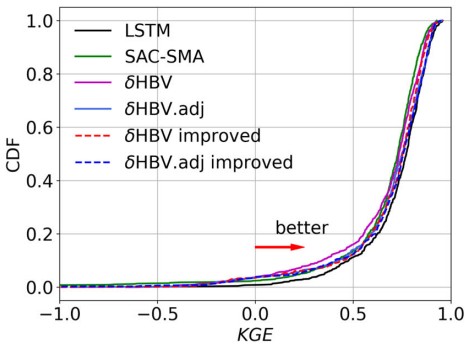





**Figure 2:  Empirical cumulative distribution function of test performance metrics for all models: Nash-Sutcliffe Efficiency (NSE, left) and Kling-Gupta Efficiency (KGE, right). LSTM represents a fully data-driven deep learning model previously used in Kratzert et al. (2019), and SAC-SMA is a purely process-based model and the simulation results are provided in CAMELS. δHBV represents the original differentiable explicit "sequential" HBV model, and δHBV.adj is the implicit adjoint-based HBV model. "Improved" indicates models where a capillary flux was added from the lower subsurface zone to the surface soil to mitigate issues with zero and low flows.**

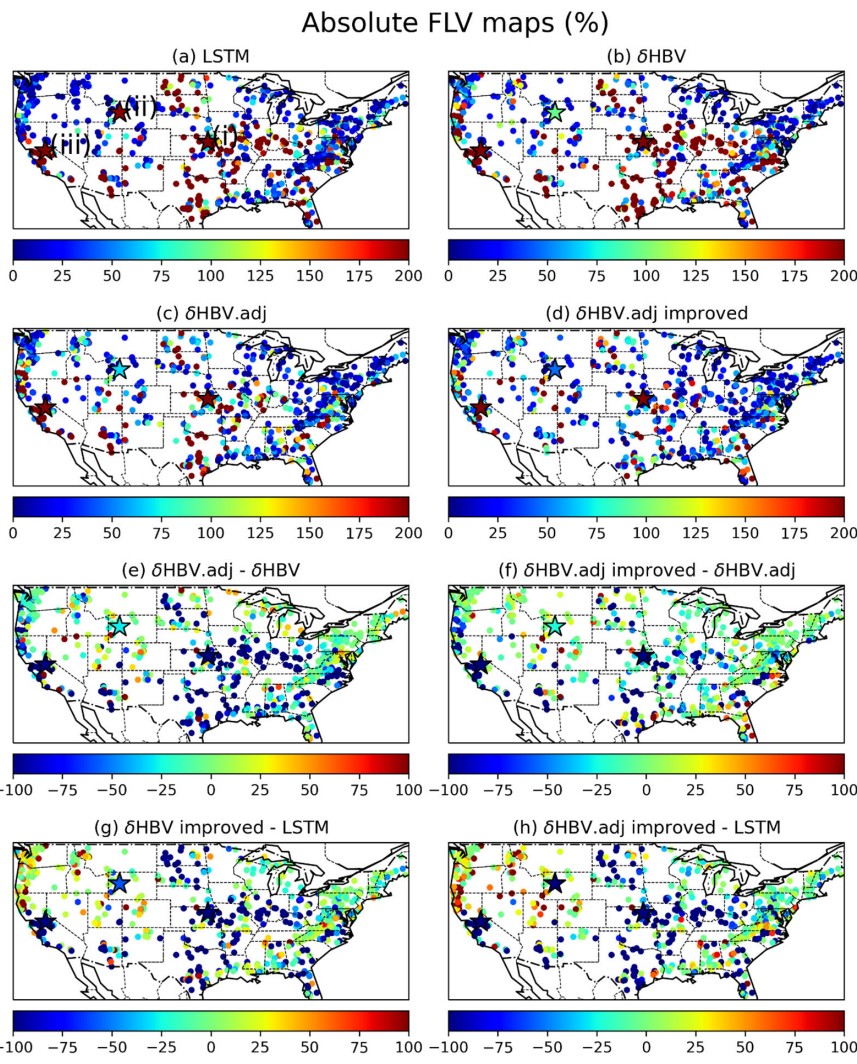


**Figure 3. Maps of percent of absolute bias of the bottom 30% ("low") flow range (absolute FLV, %) of (a) the LSTM model, (b) the sequential model (c) the adjoint model, and (d) the adjoint improved model; and maps of differences in FLV between (e) the adjoint model and the sequential model, (f) the adjoint improved model and the adjoint model, (g)the sequential improved model and the LSTM model, and (h) the adjoint improved model and the LSTM model. In (e), (f), (g), and (h), blue color indicates an improvement in baseflow representation. The sites annotated in the maps represented by star-shaped points (and labeled with i, ii, and iii in (a)), represent the locations for the plots in Figure 5.**




## 3.2 The impact of structure changes of HBV

Although the implicit scheme improved the simulations, all models, including the LSTM, exhibited significant underperformance within the geographical expanse of the Great Plains (Figure 4). This underperformance was particularly pronounced in areas marked by low, or even zero, baseflow conditions (Figure A1d in Appendix). In particular, the original HBV model encountered challenges in accurately simulating instances of zero flow due to its structural limitations, resulting in high FLV values (Figure 3b&c). Specifically, even a minimal precipitation event leads to the creation of a recharge flow in HBV from the soil moisture zone to the subsurface soil zone, which 465 subsequently contributes to the base flow. Consequently, a structural refinement of the HBV model is needed to facilitate the accurate simulation of zero flow conditions.

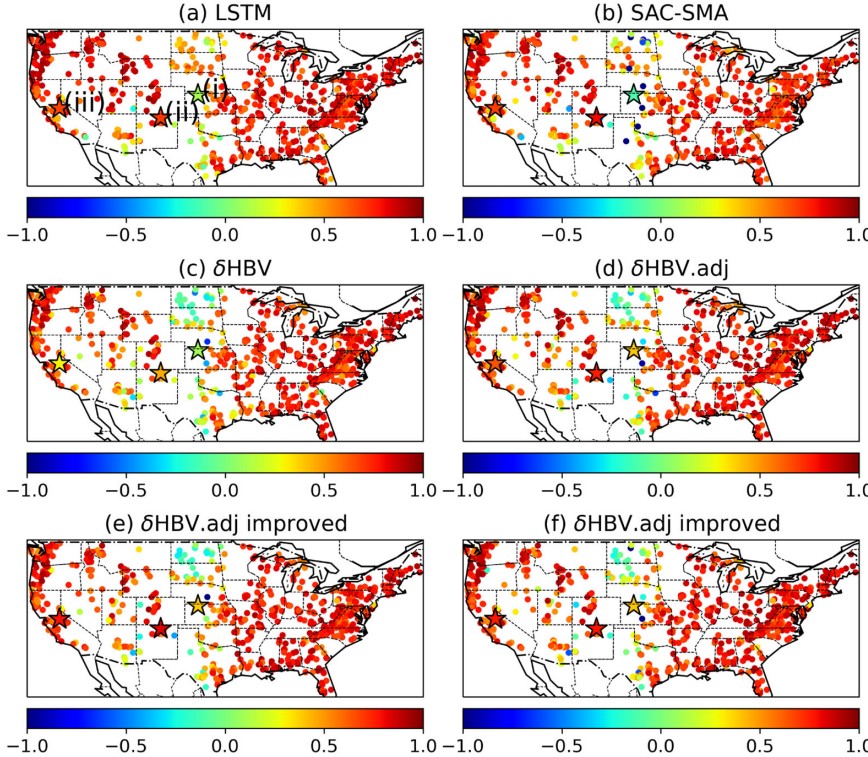

Figure 4. Maps of Kling-Gupta Efficiency (KGE) for (a) LSTM, (b) SAC-SMA, (c) The sequential model (d) the adjoint
model, (e) the sequential improved model, and (f) the adjoint improved model. The sites annotated in the maps represented by star-shaped points labeled with (i), (ii), and (iii) represent the locations for the plots in Figure 5.

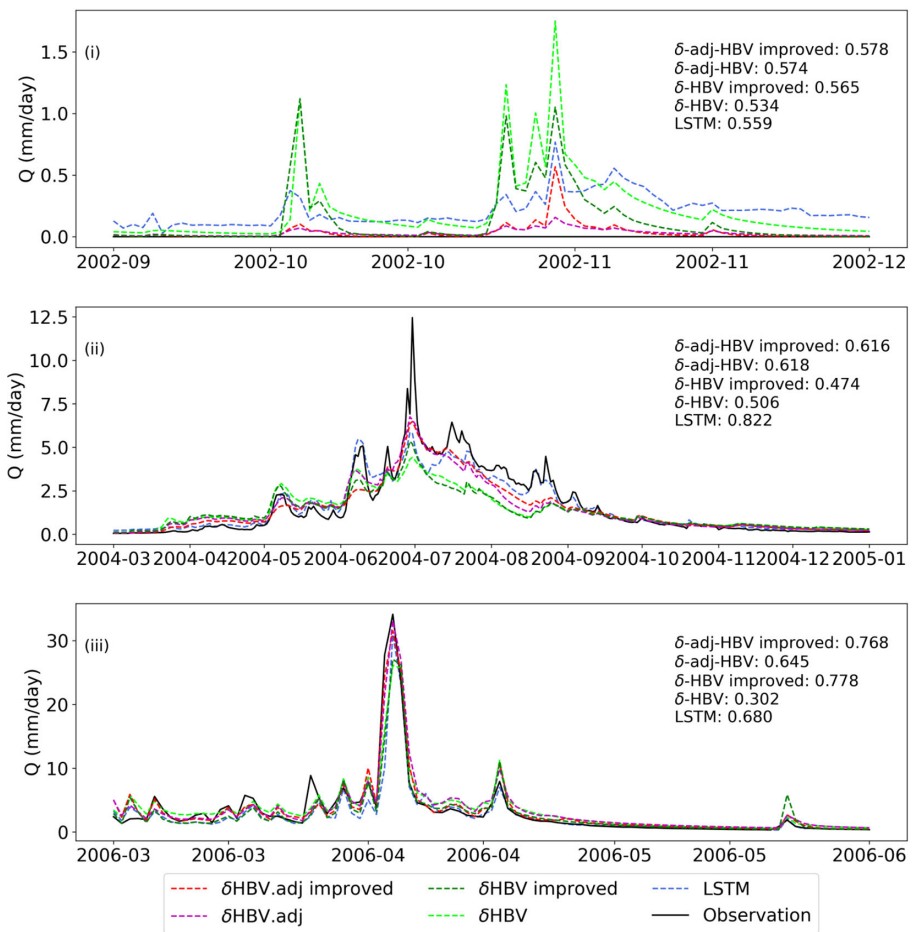

**Figure 5: The time series of streamflow simulations from models and observation at locations annotated in Figure 4 and Figure 3. KGE values of models in the whole testing period are listed in the subfigures.**

As we enhanced the HBV model by adding a capillary rise from the lower subsurface zone to the soil zone, the baseflow simulations were improved for both the sequential and adjoint models (Table 1 and Figure 3c&d). The adjoint improved model is now the most preferred model for NSE and baseflow metrics. The improvements over the default adjoint model in both absolute FLV (from 43.29 to 37.63 mm/day) and baseflow index (from 0.83 to 0.86) are noteworthy. FLV is sensitive to errors for arid regions. This means a decent description of baseflow in arid regions

needs a mechanism to help the model produce zero or near-zero baseflow, such as returning water from the lower zones to the upper zone (though multiple other structural changes may have similar effects - see Section 2.2.2), which impacts both baseflow and high flows.

The structural change mostly continued to improve FLV in regions where FLV was high and using the adjoint solver improved FLV, suggesting this module similarly gave the model more representation power to adapt to data. FLV is

substantially reduced within the western coastal regions, the south western region, the Great Plains, and the Gulf



Coastal Plain, characterized by relatively flat topography and where groundwater-driven flow contributes proportionally less to the overall streamflow dynamics, as evident from the associated low baseflow index (Figure 3f and Figure A1d).

The structural changes in $\delta$HBV models can lead to improvements with baseflows, which seems to come with a

penalty in the high flows. The best adjoint improved model has a low-flow RMSE of 0.047 mm/day, better than the sequential model, the adjoint model without capillary rise, or LSTM (Table 1). However, the high-flow RMSE did see a slight increase from 2.47 to 2.59 mm/day, but still better than the sequential models (Table 1 & Figure 5, site iii). A critical divergence in absolute FLV of $\delta$HBV improved models and LSTM appears in the center US (Figure 3g&h), where accurate flow predictions depend significantly on the fidelity of actual ET simulations.

**3.3 Analysis of impacts on parameterization**

The adjoint and sequential models achieved optimal performance with different dynamic parameterizations in this temporal test (trained and tested on the same basins but in different periods). The sequential model used $\gamma$ and $\beta$ as dynamic parameters while the adjoint model used dynamic $\gamma$, $\beta$ and field capacity, $\theta_{FC}$. $\theta_{FC}$ plays a significant role in computing effective rainfall, excess, evapotranspiration (ET), and capillary rise, exerting substantial influence over

infiltration and recharge mechanisms. Implicit schemes, involving more intricate computations like solving nonlinear equations, enables greater adaptability to data. Adapting $\theta_{FC}$ dynamically can improve the model's ability to represent real-world hydrological behavior, such as soil shrink and swell, frozen ground, soil surface sealing and expansion of the saturation excess areas (or variable source area) (Schneiderman et al., 2007) which are not directly considered in the default HBV.

The spatial patterns of the regionalized HBV parameter values demonstrate moderate differences but still a significant level of consistency between the sequential and adjoint models, suggesting that our regionalization is robust. These patterns also exhibit similarities to the parameter values estimated by Beck et al. (2016) (Figure A2 in the Appendix, reprinted) and also conform well with large-scale climate patterns on CONUS (Figure 6 a&b). It was known previously that explicit and implicit schemes arrive at very different optimal parameters (Kavetski and Clark, 2010) so we had

expected larger discrepancies, but the results show only moderate shifts. Such consistency is likely due to the strong implicit constraints imposed by parameter regionalization using data from the whole CONUS. Since all basins are served by the same neural network for the mapping, $\theta = g_w(x, A)$, it ensures autocorrelation in the parameter fields due to autocorrelation in the used predictors, and thus suppresses overfitting to local noise and numerical errors. As a result, the existence of numerical errors alone did not lead to noisy metric surface for even the explicit model (as

shown in (Kavetski and Clark, 2010)). Previously, it was difficult to efficiently impose such strong constraints and nearly optimally learn the parameters, but the differentiable modeling framework can enable regionalization at low cost and high parallel efficiency.

Delving deeper into the parameter field changes due to employing the implicit solver, we found that the adjoint model seems to have tampered down some large (close-to-bound) parameter values, which suggests that parameter

compensation for numerical error is mitigated. The shape coefficient, $\beta$, exhibits larger values (>4) within warm climate regions, while lower values (<3) characterize cold and mountainous regions (Figure 6 & Figure A1a). In the





North Dakota, Gulf Coastal Plain, and Florida, the adjoint model predicts a reduced $\beta$ compared to the sequential model. $\beta$ can influence the flashiness of the peaks and a larger $\beta$ tends to cause more threshold-like behaviors. Since the sequential model calculates ET after effective rainoff and excess, the available water for runoff is more than the

for the adjoint model which solves the equation implicitly, and thus needs such a large $\beta$. A similar pattern is observed for the field capacity, $\theta_{FC}$, of the sequential model. However, the adjoint model's field capacity estimation for the northeastern US is notably reduced compared to the sequential model, attributed to smaller clay fraction and forest fraction (Figure A1b&c), crudely aligning with estimated $\theta_{FC}$ reported by Beck et al. (2016).

The adjoint model provided a reasonable estimation for other key parameters, including the recession coefficient of

the lower subsurface zone ($\theta_{K_2}$) and the wilting point ($\theta_{LP}$). $\theta_{K_2}$ usually exhibits a correlated pattern to the baseflow index (BFI) (Figure A1d). Higher BFI indicates greater groundwater-based base flow, corresponding to a lower $\theta_{K_2}$ value that leads to diminished groundwater discharge during low-flow periods. Both The sequential and adjoint models exhibited a consistent $\theta_{K_2}$ pattern that contrasts with the baseflow index (BFI) pattern (Figure 6 & Figure A1d). Overall, the estimated wilting point, $\theta_{LP}$, of the sequential model is lower than that of the adjoint model, leading to

increased ET. As mentioned, ET being underestimated arises from the sequential solving approach of the sequential model, a smaller $\theta_{LP}$ compensate for such numerical errors.

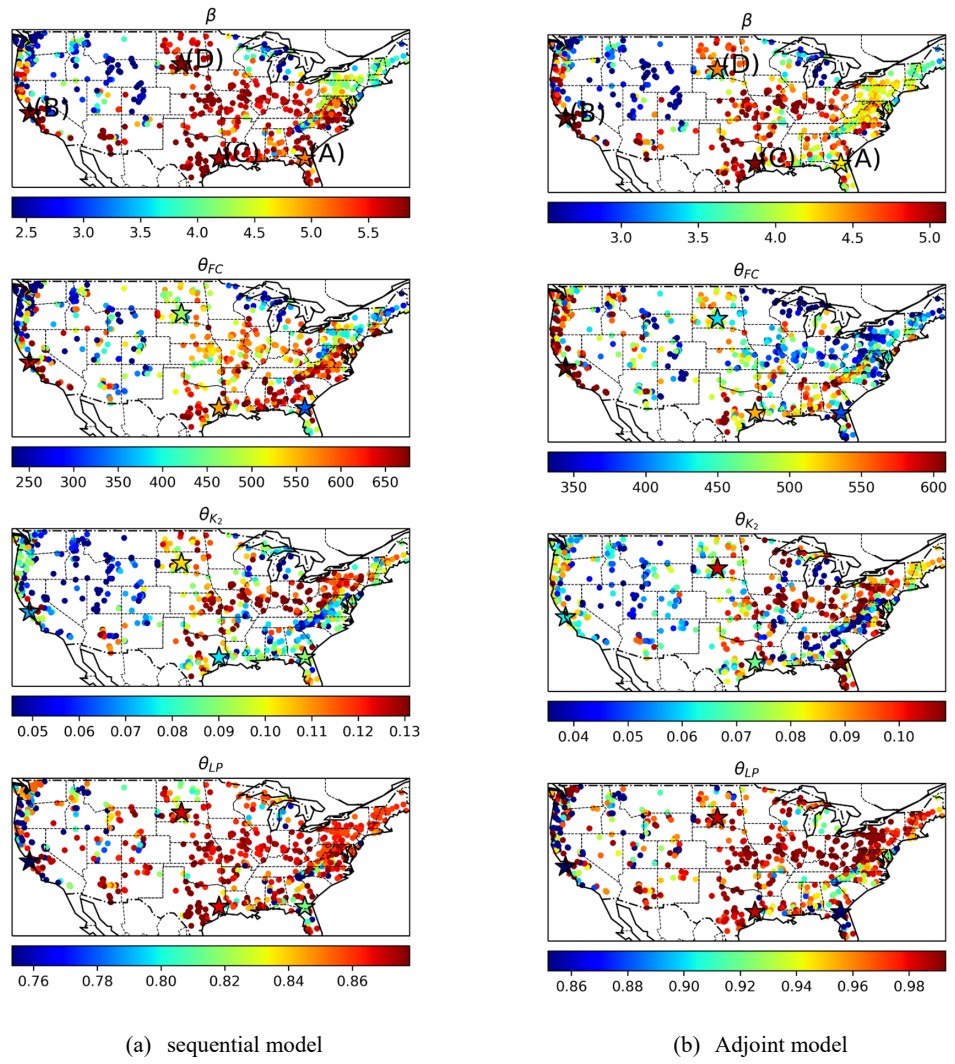

(a)  sequential model          (b)  Adjoint model

**Figure 6: Map of the optimized parameter β, field capacity, $\theta_{FC}$, the recession coefficient of lower subsurface zone, $\theta_{K_2}$, and the wilting point, $\theta_{LP}$, from (a) sequential model and (b) adjoint model. The sites annotated in the maps represented by star-shaped points labeled with letters A-D, represent the locations for the plots in Figure 7.**

The sequential and adjoint models exhibit similar sensitivity patterns across varying HBV parameters in high-performance basins (as illustrated in Figure 7A&B, and geographical locations in Figure 6). This consistency again highlights the stability of the regionalization scheme, in contrast to the noisy metric surface shown in Kavetski and Clark (2010) for single basins. For these two basins, we see a smooth contour where the NN-predicted values (annotated by the symbol "o") were not too far from the optimal value. However, in basins exhibiting poor performance (as depicted in Figure 7C&D), their differences enlarged and apparently numerical errors shifted the

parameter distributions. Although the overall contour patterns stay similar, the values of the contours have changed

quite significantly, due to the dependence on the calculation order.

The process of parameter regionalization introduces a trade-off between performance and spatial coherence, leading to parameters that might not be optimal for each specific basin; when this gap is too large, it suggests there might be some structural issues or missing information. Take basin (D) (Figure 7D) as an example, where the optimal values for $\theta_{FC}$ and $\beta$ fall within the ranges of 100 - 200 mm and 1.0 - 4.0, respectively, but the NN-predicted parameter

values (centers of the KGE contours in Figure 7) deviate significantly from these optimal ranges. The regionalized parameters thus produced a rather low KGE of -0.3. This trade-off could mean that some key processes are not well represented and the parameters could potentially compensate for these processes, but the compensation was prevented by regionalization. A notable example is the absence of topographic information and subbasin-scale spatial heterogeneity, which are crucial for modeling arid basins but were not fully considered by the present parameterization

network. These parameter gaps give us hints for the next stage of model improvements.

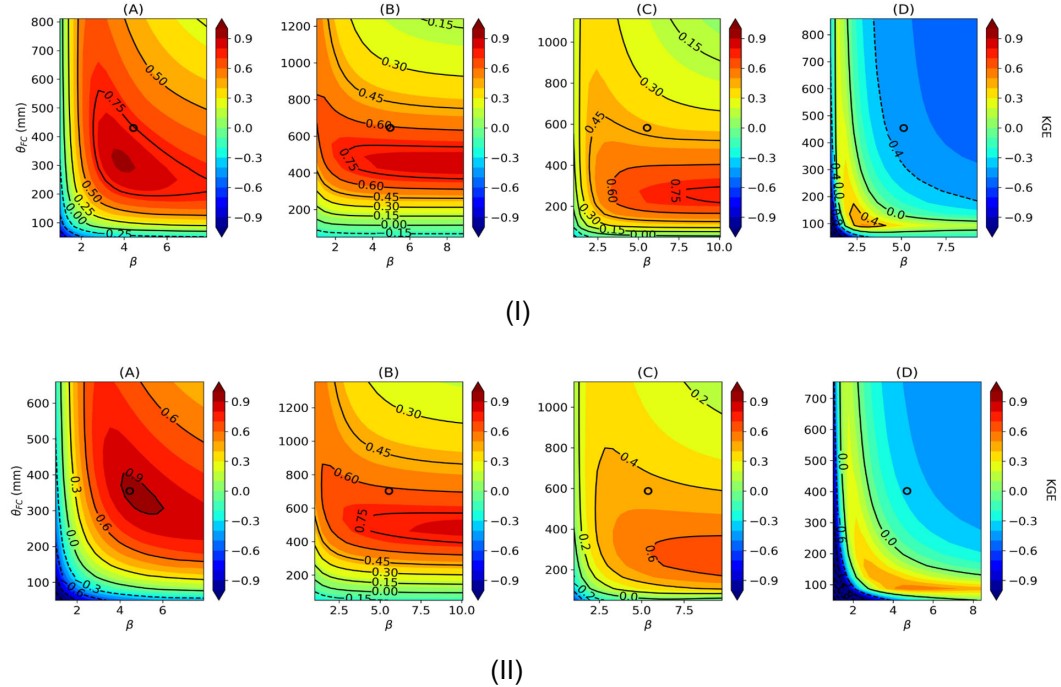

**Figure 7: Impact of numerical schemes on the KGE surface of HBV model: The contour of KGE calculated from the (I) sequential model and (II) adjoint model on the 2D slice of field capacity ($\theta_{FC}$) and parameter β. The predicted parameter values are positioned at the central point of the contours delineated by circles. The locations of selected sites are annotated**
**in Figure 6**



## 4. Further discussion

The adjoint method was used in this work to support the implicit numerical scheme in differentiable models, which allows for the efficient joint training of the neural networks with physical models using gradient descent. Such joint and "online" training on big data is not possible without the process-based model being differentiable, because the only presently known way to train such a large number of weights is via backpropagation. While we utilized a coupled neural network in this work for regional parameterization, it can also be used to replace any component of the physical model for knowledge discovery. For example, the runoff module in the HBV model could be replaced with a neural network. Thanks to their ability to train on big data, differentiable models can be trained on all basins and constrain a common neural network to learn a universal relationship between the inputs and the physical parameters. Such relationships can be used for interpolation and extrapolation at sites lacking observations.

While the differentiable implicit model outperforms the sequential one and offers state-of-the-art performance, it incurs a substantially larger computational cost. Newton's method to solve the implicit equation requires a number of iterations (Eq. 29) and, for the sake of the adjoint, we need to solve a matrix (Eq. 33). In addition, the more complicated computational instructions of the adjoint procedure may encounter higher CPU overhead and thus lower GPU utilization rate compared to the forward simulations. Due to all these reasons, the computational cost is almost 5-10 times that of the sequential model. For context, compared to running traditional models on CPUs, our implementation is already highly efficient (running the model forward over 671 CAMELS basins for 15 years takes 18 and 133 hours for the sequential and implicit models, respectively). Nonetheless, the increased computational demand for the implicit solver still creates challenges for training at large scales as it requires thousands of forward simulations. As a potential solution, based on the parameter consistency between the sequential and the implicit models, it seems we can use the inexpensive sequential model as an "explorer model" for model structure identification, hyperparameter tuning, and neural network pretraining. Then we can fine-tune the network using the adjoint model. Hence, both models offer utility for global-scale applications.

Although the adjoint models (both with and without structural changes) already outperformed LSTM in terms of low-flow and high-flow RMSE, which is an astounding result, they still slightly fell behind LSTM at certain basins with poor performance in the center and western US (arid regions). In the future, we can assess multiple hypotheses that may explain why the adjoint model's performance in these regions is not as good as that of LSTM: (A) the differentiable model is more hampered by precipitation bias in these regions than LSTM, which can internally account for predictable bias; (B) HBV's baseflow inadequacy arises due to not fully utilizing forcing information. LSTM can fully utilize information in the inputs, e.g., solar radiation and vapor pressure, while the present HBV model only uses temperature in determining PET, apart from precipitation. The difference in solar radiation and vapor pressure could have impacted long-term water balance and baseflow. Future work could use the Penman-Monteith equation for PET, which considers vapor pressure, or learn better PET equations from data (Zhao et al., 2019). (C) HBV faces a larger tradeoff between matching the high and low flow portions of the observed hydrographs, and the adjoint models sacrificed low-flow performance to some extent in favor of better overall performance; (D) the model backbone, HBV, is unable to represent some groundwater dynamics, e.g., lateral redistribution of moisture from hillslope to valley (Clark et al., 2015; Fan et al., 2019). While LSTM could internally form a cascade of neurons that transfers mass akin





to lateral groundwater movements, the two-layer groundwater structure in HBV is too simple to represent such impacts. These hypotheses will lead us to improved model structures with the help of data. An unprecedented advantage with differentiable modeling is that we can simultaneously learn robust relationships from large data and find nearly optimal parameterization schemes. This reduces the complex, iterative testing between different model structures and parameter optimization.

In future research, it would be valuable to evaluate the advantages and disadvantages of "optimize-then-discrete" versus "discretize-then-optimize" adjoint methods in hydrological differentiable models. Generally, the "optimize-then-discrete" approach tends to be more computationally demanding and less accurate because of gradient inaccuracies when the adjoint state differential equations aren't sufficiently resolved. On the other hand, the adjoint in the "discretize-then-optimize" method is solved from the Jacobian matrix using automatic differentiation, offering greater efficiency and accuracy compared to numerically solving the adjoint differential equation (Onken and Ruthotto, 2020). Nevertheless, it requires more work to compare the two options.

AD is a tool that we seldom used prior to the prevalence of machine learning. Neither did we have modern GPUs or the software to maximize its utilization. The past few years saw substantial software and hardware investments from the artificial intelligence community that have made these tools orders-of-magnitude more efficient. Utilizing these tools and running hydrologic models on such platforms means the water community can leverage these investments and can grow with the AI community at little cost. For example, the model can automatically become even faster with slight effort to embrace just-in-time compilation of torch 2.0 (Wu, 2023).

## 5. Conclusions

Our comparisons show that the numerical errors associated with the sequential model, and especially its dependence on the order of computation, had detrimental impact on its representation power -- it cannot provide high-quality low

flow, high flow, and groundwater simulations and can introduce parameter compensations. The adjoint method for gradient calculation enables the use of implicit solvers in differentiable modeling, partially mitigating the numerical errors. While not explicitly demonstrated, other hydrologic problems that require implicit solvers can similarly benefit from the adjoint method. With the implicit solver and with a structural change (capillary rise), our model comprehensively improved the simulations of low flow, and an uncalibrated variable, baseflow fraction. While some

of the differences in metrics may not seem large, they are already significant and could result in flood peaks being more accurately predicted. The comparison of baseflow simulations also implies that the same numerical issue may hamper other models so that, in order to achieve top-of-the-line performance with differentiable models, numerical errors have to be examined and the implicit model and adjoint will be needed.

The capacity of differentiable models to outperform the LSTM in low-flow and high-flow metrics at the median of

CAMELS basins proves that structural priors (and physical interpretability) and state-of-the-art performance are not mutually exclusive, and pure deep networks are not necessarily the performance ceiling our environmental models (although we do expect them to be close to optimal). In fact, for rarely-observed events, structural priors may potentially overcome data limitations. The fact that modifying the structure can result in better physical representations hints that we can make further improvements to baseflow and peak flow, and identify better structure from data.



The regionalization scheme produced overall stable parameter fields that, on first look, have similar patterns between sequential and implicit models, but a deeper investigation shows that the implicit scheme reduced large, near-bound parameter values where competition between fluxes is likely to occur. This is visual evidence that parameter compensation occurs more strongly with the sequential model and it can be mitigated. Since it is preferable to remove the interference of numerical errors prior to interpreting the parameter fields, the implicit model would be favored when the interest is in the intermediate parameters or internal fluxes. The ancient demon of numerical errors remains relevant in the new era of big data, but may be mitigated by the adaptive capability of deep networks.

**Competing Interests**

Kathryn Lawson and Chaopeng Shen have financial interests in HydroSapient, Inc., a company which could potentially benefit from the results of this research. This interest has been reviewed by The Pennsylvania State University in accordance with its individual conflict of interest policy for the purpose of maintaining the objectivity and the integrity of research. The other authors have no other competing interests to declare.

**Acknowledgments**

YS and CP were supported primarily by subaward A22-0307-S003 from Cooperative Institute for Research to Operations in Hydrology (CIROH) through the NOAA Cooperative Agreement (grant no. NA22NWS4320003) and partially by US Department of Energy under award DE-SC0016605. WK and MC were also supported by CIROH. DF was supported by US National Science Foundation (NSF) EAR-2221880. In addition, the authors wish to thank Kamlesh Sawadekar for providing LSTM results. Computational resources have been partially provided by the NSF OAC-1940190.

**Data Availability Statement**

The code for the explicit differentiable models are available at https://doi.org/10.5281/zenodo.7091334. A new code release, together with a software update, will be produced upon paper acceptance. The code for the LSTM model can be downloaded at GitHub (https://github.com/mhpi/hydroDL). The CAMELS dataset can be downloaded at https://dx.doi.org/10.5065/D6MW2F4D (Addor et al., 2017; Newman et al., 2014). MODIS ET data can be downloaded at https://modis.gsfc.nasa.gov/data/dataprod/mod16.php (Running et al., 2017).

**Appendix**

Table A1 are the forcing and attribute variables used in the LSTM models.

**Table A1: Summary of the forcing and attribute variables used in all Models**





|  | Variable | name | Unit |
|---|---|---|---|
| Forcings | PRCP | Precipitation | mm/day |
|  | $E_p$ | Potential evapotranspiration | mm/day |
|  | T | Temperature | °C |
| Attributes | p_mean | Mean daily precipitation | mm/day |
|  | pet_mean | Mean daily PET | mm/day |
|  | p_seasonality | Seasonality and timing of precipitation | - |
|  | frac_snow | Fraction of precipitation falling as snow | - |
|  | Aridity | PET/P | - |
|  | high_prec_freq | Frequency of high precipitation days | days/year |
|  | high_prec_dur | Average duration of high precipitation events | days |
|  | low_prec_freq | Frequency of dry days | days/year |
|  | low_prec_dur | Average duration of dry periods | days |
|  | elev_mean | Catchment mean elevation | m |
|  | slope_mean | Catchment mean slope | m/km |
|  | area_gages2 | Catchment area (GAGESII estimate) | $km^2$ |
|  | frac_forest | Forest fraction | - |
|  | lai_max | Maximum monthly mean of the leaf area index | - |
|  | lai_diff | Difference between the maximum and minimum monthly mean of the leaf area index | - |
|  | gvf_max | Maximum monthly mean of the green vegetation | - |
|  | gvf_diff | Difference between the | - |





| | | | |
|---|---|---|---|
| | | maximum and minimum monthly mean of the green vegetation fraction | |
| | dom_land_cover_frac | Fraction of the catchment area associated with the dominant land cover | - |
| | dom_land_cover | Dominant land cover type | - |
| | root_depth_50 | Root depth at 50th percentile, extracted from a root depth distribution based on the International Geosphere-Biosphere Programme (IGBP) land cover | m |
| | soil_depth_pelletier | Depth to bedrock | - |
| | soil_depth_statgso | Soil depth | m |
| | soil_porosity | Volumetric soil porosity soil_conductivity | - |
| | soil_conductivity | Saturated hydraulic conductivity | cm/hr |
| | max_water_content | Maximum water content | m |
| | sand_frac | Sand fraction | - |
| | silt_frac | Silt fraction | - |
| | clay_frac | Clay fraction | - |
| | geol_class_1st | Most common geologic class in the catchment basin | - |
| | geol_class_1st_frac | Fraction of the catchment area associated with its most common geologic class | - |
| | geol_class_2nd | Second most common geologic class in the catchment basin | - |
| | geol_class_2nd_frac | Fraction of the catchment area associated with its 2nd most common geologic | - |





| | | class | |
|---|---|---|---|
| | carbonate_rocks_frac | Fraction of the catchment area as carbonate sedimentary rocks | - |
| | geol_porosity | Subsurface porosity | - |
| | geol_permeability | Subsurface permeability | $m^2$ |


We provided the maps four attributes, aridity, forest fraction, caly fraction, and the baseflow index, used in the differential models to support our analysis.

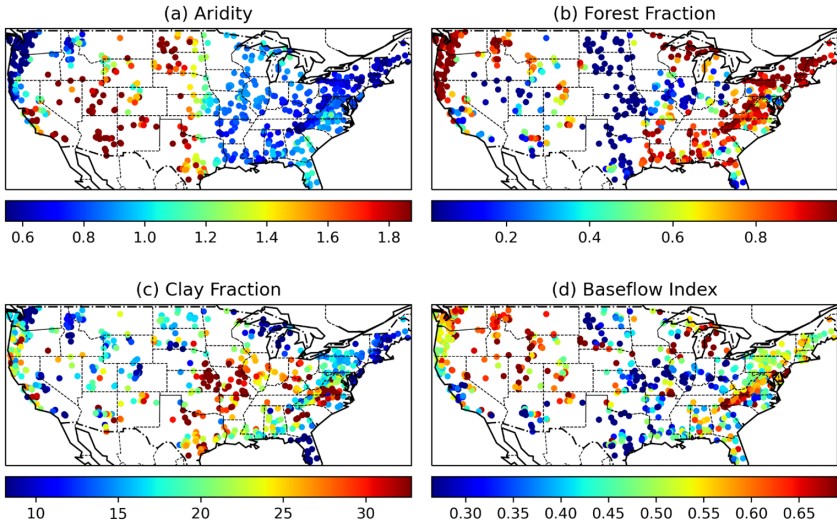

**Figure A1: Map of the static attributes: (a) Aridity, (b) Forest Fraction, (c) Caly Fraction, and (d ) Baseflow index from CAMELS dataset.**

We reprint the Figure 4 in Beck et al. (2016) to facilitate a comprehensive parameter comparison.


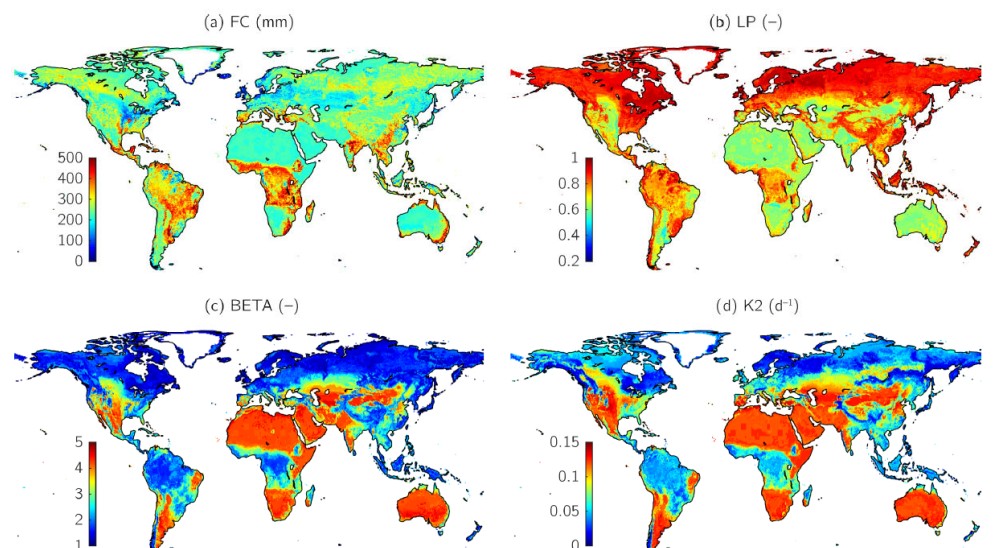

**Figure 4.** For HBV, mean values of the regionalized parameters based on the 10 most similar donor catchments for: (a) the maximum water storage in the unsaturated-zone store (FC); (b) the soil moisture value above which actual evaporation reaches potential evaporation (LP); (c) the shape coefficient of recharge function (BETA); and (d) the recession coefficient of lower groundwater store (K2). For maps of all other parameters, see supporting information Figure S2.1.

**Figure A2: For HBV, mean values of the regionalized parameters based on the 10 most similar donor catchments for: (a) the maximum water storage in the unsaturated-zone store (FC); (b) the soil moisture value above which actual evaporation reaches potential evaporation (LP); (c) the shape coefficient of recharge function (BETA); and (d) the recession coefficient of lower groundwater store (K2). Reprinted with permission from Beck et al. (2016).**

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
