# Peer review of "When ancient numerical demons meet physics-informed machine learning: adjoint-based gradients for implicit differentiable modeling"

_Hydrology and Earth System Sciences, 2023_

## Referee Comment (RC2)

**Review of Manuscript**

**'When ancient numerical demons meet physics-informed machine learning: adjoint-based gradients for implicit differentiable modeling'**

By Y. Song et al.

Dear Editor,

I have reviewed the aforementioned work. My conclusions and comments are as follows:

**1. Scope**

The article is within the scope of HESS.

**2. Summary**

In their paper, the authors introduce an adjoint-based method to allow for efficient training of hydrological models via backpropagation, even if they use implicit numerical solvers. At first, the authors give an overview on the current state of hydrological modeling with a focus on the different approaches (conceptual modeling based on physical process understanding, purely data-based such as LSTMs, and hybrid combinations thereof) and on how formulating these models in a differentiable manner is key for their efficient training through backpropagation. They also point out that typical physics-based conceptual hydrological models rely on explicit, non-iterative numerical schemes, which potentially introduces numerical error and hampers efficient model training and unambiguous parameter identification. In this context, the main goals of the paper are therefore to **i)** introduce the adjoint-based method ('discretize-then-optimize'), **ii)** compare and evaluate models with implicit and explicit numerical schemes, and **iii)** to assess whether local and regional parameter distributions differ between the two schemes.

The authors use the well-known CAMELS-US data set to set up and compare various models: An LSTM-only and the SAC-SMA model as benchmarks, and several variants of a hybrid LSTM-HBV model, where the LSTM provides the parameterization for the HBV model. The variants comprise the four possible combinations of two numerical solvers (explicit and implicit), and two model architectures (without/with capillary rise).

With respect to i) and ii), the authors compare the LSTM-only and the LSTM-HBV hybrids with explicit and implicit numerical schemes for a range of performance measures and conclude that the implicit model improves streamflow simulation (lines 412 pp) and that this can be attributed to the numerical errors of the explicit numerical scheme (lines 413-415).

The authors then discuss the effects of the changed HBV model structure (addition of capillary rise) to help the model produce (near-)zero base flow in accordance with observations, and conclude that the added process is helpful in low-flows, but comes with some deterioration for high-flows.

With respect to iii), the authors compare spatial parameter patterns of the HBV hybrids with different numerical schemes, and conclude that the patterns largely agree, indicating the robustness of the involved parameter regionalization scheme.

**3. Evaluation**

First of all, I acknowledge the work of the authors to further merge data- and physics-based approaches to modeling by showing how models using implicit solvers can be integrated into a typical machine-learning workflow with backpropagation at its core. This is a valuable contribution to the hydrological modeling sciences, but unfortunately the authors do not convincingly prove in their

paper the immediate benefit thereof, and they obscure their point by adding aspects to the study that are not related to the main message. I will explain this in the following:

A) The authors are correct that mainstream conceptual hydrological models like HBV have traditionally - and often without much reflection – been used with simple explicit numerical schemes, and a pre-set order of process execution, and that this may cause substantial problems (see Clark and Kavetski, 2010 as cited by the authors), and that implicit schemes can solve these problems. Therefore, in this manuscript, in addition to the description of how to include implicit schemes in ML-workflows, I was expecting a demonstration of how this actually solves a problem. That is, showing that for a particular hydrological modeling task (here: modeling streamflow in daily resolution of the CAMELS-US basins) i) the standard explicit scheme introduces problems and ii) that an implicit scheme solves them. The authors mention this point in the paper (line 324-326), but unfortunately do not address it. For example, one could operate HBV models for some of the CAMELS catchments with various execution orders and extremely fine-grained time-stepping, thus effectively removing the detrimental effect of the explicit scheme, and then compare to a standard time stepping and execution order, and to a model using an implicit scheme. The authors conclude in their study that the (small) model improvements between the HVB-hybrid variants using explicit and implicit schemes are due to problems introduced by the explicit scheme (lines 412-415), but because they do not provide a proof for a cause, the conclusion based on an effect is not convincing.

In this context, it might also be interesting to analyze if decreasing negative effects of explicit schemes by higher time stepping (or other changes to the model computational setup) might be more efficient than shifting to implicit schemes. The authors mention that computational costs for the latter increased by a factor of 5-10 (line 581). Increasing the time stepping of the explicit scheme from daily to 6 hours would only mean a factor of 4, but would already resolve diurnal cycles, which might be relevant additional information for the model.

B) Motivated by problems of the HBV model to simulate (near-)zero base flow during extended dry spells, the authors integrate a detailed study about the effect of adding an additional capillary rise process to the HBV model. This is a valid question and analysis, but it does not at all support the main argument of the paper about how and why implicit schemes can be integrated into modern hybrid modeling workflows. I therefore suggest presenting this analysis in another paper, and removing it from this one.

In this context, it is interesting that the authors provide a range of possible adjustments to the HBV model to help it achieve (near-)zero flow (strategies 1-5 in lines 258-261; and lines 479-482). These adjustments touch very different physical subdomains and processes of the model, and one may wonder about the limitations of a supposed key advantage of physics-based models – realism and interpretability – if it remains mainly up to the user's preference which one is chosen. In particular, I wonder why capillary rise from the lower subsurface should bypass the upper surface and directly connect to the surface soil moisture storage, and why the authors chose it this way.

Based on the above points, my overall recommendation is that the key topic of the paper is worth publication, but also that the required changes will require time. Therefore I recommend rejecting the paper in present form, but strongly encourage a resubmission.

Yours sincerely,

Uwe Ehret

---

## Referee Comment (RC3)

**Replies to author comments AC2 and AC3 by Chaopeng Shen for manuscript**

**'When ancient numerical demons meet physics-informed machine learning: adjoint-based gradients for implicit differentiable modeling'**

By Y. Song et al.

Dear Chaopeng Shen, dear Authors, dear Editor,

Thanks for your detailed replies to my review. This is exactly what the HESS open discussion forum is made for. First of all, I would like to state once more that I think the authors, by introducing in their manuscript a method for integrating implicit solvers in modern ML-based model training via backpropagation, provide valuable research that is absolutely worth publishing. The reason I was (and am) recommending "reject with strong encouragement for resubmission" is that I suggest changes to the manuscript that will most likely require more time than usually assigned for major revisions. I will leave it to the Editor whether he thinks i) the changes I suggest are valid, and if yes, ii) how much time they would require.

Reading the replies (AC2, text and .pdf, and AC3) by Chaopeng Shen, four main points of discussion arise. I reply to them here in summarized form, rather than individually in each document:

- **The main purpose of the paper is to enable implicit schemes**. I agree with this statement by Chaopeng Shen, and I suggest focusing on this message. Therefore, I would support a revised paper, as a technical note, that introduces the method, not more and not less. In such a paper, there is no need for an in-depth discussion about if and when explicit schemes for conceptual hydrological models operating on daily data create problems, and no need for comparing implicit schemes vs. explicit schemes operated on higher-resolution data comparison.

- **Running small time steps (less than a day) with automatic differentiation creates problems (memory use, allowable window size)**. Chaopeng Shen suggests it is a bit unfair by me to ask the authors to demonstrate that implicit schemes solve a problem of explicit schemes. I see two options here. The first is to write a short technical note presenting only the main method innovation, see previous bullet point. The second is to keep it as a research paper, showing the method innovation and applications. In that case, as a reader I would expect a demonstration that the innovation solves an important problem present in the applications used in the paper (hydrological modeling on daily basis using conceptual models). That is, showing that the currently used explicit schemes introduce substantial numerical error. This does not need to be for the full set of catchments used, but could be done for a few representative catchments, and along the lines of the demonstrations in Clark and Kavetski (2010). If the editor thinks this is unnecessary detail, I would at least expect a more in-depth discussion about how the findings of Clark and Kavetski (2010) and Kavetski and Clark (2010) apply to the application in the manuscript.

- **Adaptive time stepping is difficult to realize in connection with AD, more specific in connection with parallel processing of minibatch optimization.** I never tested, but Chaopeng Chens explanation of this point makes sense to me. In my review, I never asked the authors to include such tests in the manuscript, therefore there is no disagreement here.

- **Discussion of HBV structural/functional changes (capillary rise) in the same paper**. In my review, I was mentioning that discussing structural/functional changes to the HBV model to solve an apparent model deficiency has little to do with the key message of the manuscript, and therefore suggested removing it. I still think that leaving this part away will help the paper to better convey its message. The reply by Chaopeng Shen - "Many article carry more than one stories and this is a beneficial (although not that major) improvements to the model. We do not want to write another article for this change." - has not convinced me otherwise. I will leave this decision to the editor.

Yours sincerely, Uwe Ehret

---

## Author Comment (AC1)

*Dear authors, Dear editor,*

*Here is my review of the submitted work. I recommend accepting the manuscript after minor revisions.*

*Kindly*

*Ilhan Özgen-Xian*

*General comments and questions*

*1. The authors convincingly make an argument for implicit time integration. The forward Euler time stepping used in this work is indeed at a disadvantage if fixed time steps are used. However, it is not clear to me how higher order explicit time integration methods such as schemes from the explicit Runge-Kutta family (RK) would perform in comparison to the implicit one. If I understood correctly, some of the numerical issues mentioned in the manuscript might also be addressed by (adaptive) multistep schemes of this type. The advantage of RK-type schemes in this context is that the number of computations per time step is known a priori. In contrast, the Newton-Raphson iterative solver may require any number of steps until convergence. High order RK schemes, for example the standard RK45 or the adaptive RK-Fehlberg method, could also potentially benefit from the adjoint method presented in this paper to avoid excessive memory usage. Perhaps the authors can comment on this.*
*2. The authors mention that the Newton-Raphson solver introduces some overhead to the computation. On average, in the results shown in this paper, how many iteration steps were necessary for the solver to converge?*

Thanks for your suggestions. We think these two questions can be answered together. The Newton-Raphson solver can converge in 3-4 iterations on average (We added this information in the main text). Its computational cost actually is less than the high-order explicit Runge-Kutta method, such as 4th order Runge–Kutta–Fehlberg method. RK methods could require more memory usage during the backpropagation process for gradient calculation because every step of the calculation needs to record information and store intermediate data. In standard ODE solvers and during our tests on chaotic ODE problems, RK methods sometimes run into stiff or stability issues and need to reduce time steps or change to lower-order methods, this further increases memory use and challenges to parallel efficiency. Thus, while some high-order RK methods can be useful (and we think they can be a list of options provided), they also have risks. In addition, the disadvantage of the explicit/implicit iterative solvers is not only the memory usage but also the gradient vanishing or explosion due to the gradient accumulation over all the time steps and iterations in the training instance. The adjoint for the implicit iterative solver can bypass the gradient tracking in the iterations of each time step.

Here, we do not argue that implicit solutions are the only way. We think that explicit solutions can in some cases be useful. Nevertheless, implicit methods are well known to provide important value for various problems so they must be provided as an option to differentiable models. For example, elliptic problems must be handled by implicit schemes and stiff ODEs are best handled by them, too. We will add some explanations: "*While this*

*paper focuses on enabling implicit solvers in differentiable modeling, we do not suggest that explicit solvers are to be discouraged. Runge-Kutta schemes can be well suited for a number of cases and may be attempted for the rainfall-runoff case. It has long been explored in the numerical algorithm literature that each type of solvers has their advantages and disadvantages and is suitable for different problems. For example, implicit solvers are not only preferred but also necessary for stiff ODEs, especially those with dynamics on vastly different time scales and those resulting from the discretization of elliptic PDEs. Using explicit solvers for them could necessitate very small time steps which need to be coordinated with the modification of forcing inputs. In the context of differentiable modeling, a new dimension of consideration plays an important role --- GPU parallel efficiency at the batch level --- because the primary point of differentiable modeling is to learn from big data. Either explicit or implicit scheme needs to serve this purpose. This means that time-adaptive solvers that may require vastly different time steps amongst batch members may have limited applicability when we want to use minibatches. In addition, as discussed in the Introduction, all automatic differentiation steps incurs CPU overhead and storage burdens --- thresholds and array mutation, especially, often require data storage on the GPU. GPU memory may soon run out if we have too many iterations, either with explicit or implicit schemes, which could limit the training lengths. If neural network weights participate in the calculations of these iterations, it further induces the problem of vanishing gradients. We need to put these constraints into consideration and design balanced algorithms.*"

*Minor comments*

*1. P.2, L.70: "graphical processing units" should be "graphics processing units"*

Will revise as suggested.

*2. P.3, LL.105ff.: Does "elliptic operator" in this context correspond to the Laplacian?  If so, some of the examples  might require some annotation.  The Saint-Venant equation only contains Laplacian operators if molecular/turbulent diffusion is accounted for.  Many forms of the Saint-Venant equation omit these terms, for example (García-Navarro et al., 2019, doi:10.1007/s10652-018-09657-7; LeVeque et al., 2011, doi:10.1017/S0962492911000043).*

We will revise it to "shallow water equations", which is a two-dimensional Saint-Venant equation considering turbulent diffusion.

*3. P.3, LL.105ff. (continued) When I looked at the paper by Aboelyazeed et al. (2023) (cited by the authors), I couldn't see Laplacians in the Farquhar model equations.*

Aboelyazeed et al. (2023) is an example of systems of nonlinear equations, not an elliptic operator.

*4. P6, L.209: "The same forcings ... was used" should be "The same forcings ... were used"*

Will revise as suggested.

*5. P.12, L.335: Should it be Eq. (28) instead of Eq. (27)? May be I am misunderstanding something.*

Yes, your understanding is correct. We will fix it.

*6. P.14, L.398: The authors state that the mass balance preservation of the adjoint-driven NN-HBV model might be the reason behind the improved model performance. I don't understand why the mass conservation should significantly differ from the explicit sequential NN-HBV model if the hydrological process representation remains untouched. Is this related to the use of thresholds to avoid negative storages? Can the authors elaborate a bit more?*

Yes, by avoiding thresholds for negative states, the implicit model can achieve better mass conservation. This impact is significant for low flows. For example, the thresholds for lower subsurface zone storage in the current HBV model can induce minimal baseflow on dry days. More importantly, the adjoint (implicit) model greatly reduces numerical errors, thus improving the model's performance. We revised the main text to: *"The advance may be attributable to HBV.adj's reduction of numerical errors, which forces the model to more accurately represent extreme values."*

*7. P.24, L.580: The additional computational cost introduced by the implicit solver is quite substantial (18 h vs. 133 h), suggesting either poor convergence or large communication overhead in the implicit scheme.*

We agree with the reviewer that the computational cost of the implicit solver is substantial compared to the sequential model. However, when compared with traditional models that require basin-by-basin calibration on a CPU, it is efficient for large-scale modeling. The implicit solver can converge in 3-4 iterations, and all training is conducted on a single GPU, no need for communication between nodes. The reasons it is slower than the sequential model are twofold: 1) the HBV model is called 3-4 times in each time step, whereas the sequential model only needs to call it once, and 2) the calculation of the Jacobian matrix for multiple basins, depending on the batch size, also consumes time. There are additionally some CPU overhead issues to be explored down the road.

---

## Author Comment (AC2)

*Reviewer #2*

*First of all, I acknowledge the work of the authors to further merge data- and physics-based approaches to modeling by showing how models using implicit solvers can be integrated into a typical machine-learning workflow with backpropagation at its core. This is a valuable contribution to the hydrological modeling sciences, but unfortunately the authors do not convincingly prove in their paper the immediate benefit thereof, and they obscure their point by adding aspects to the study that are not related to the main message. I will explain this in the following:*

*A) The authors are correct that mainstream conceptual hydrological models like HBV have traditionally - and often without much reflection – been used with simple explicit numerical schemes, and a pre-set order of process execution, and that this may cause substantial problems (see Clark and Kavetski, 2010 as cited by the authors), and that implicit schemes can solve these problems. Therefore, in this manuscript, in addition to the description of how to include implicit schemes in MLworkflows, I was expecting a demonstration of how this actually solves a problem. That is, showing that for a particular hydrological modeling task (here: modeling streamflow in daily resolution of the CAMELS-US basins) i) the standard explicit scheme introduces problems and ii) that an implicit scheme solves them. The authors mention this point in the paper (line 324-326), but unfortunately do not address it. For example, one could operate HBV models for some of the CAMELS catchments with various execution orders and extremely fine-grained time-stepping, thus effectively removing the detrimental effect of the explicit scheme, and then compare to a standard time stepping and execution order, and to a model using an implicit scheme. The authors conclude in their study that the (small) model improvements between the HVB-hybrid variants using explicit and implicit schemes are due to problems introduced by the explicit scheme (lines 412-415), but because they do not provide a proof for a cause, the conclusion based on an effect is not convincing. In this context, it might also be interesting to analyze if decreasing negative effects of explicit schemes by higher time stepping (or other changes to the model computational setup) might be more efficient than shifting to implicit schemes. The authors mention that computational costs for the latter increased by a factor of 5-10 (line 581). Increasing the time stepping of the explicit scheme from daily to 6 hours would only mean a factor of 4, but would already resolve diurnal cycles, which might be relevant additional information for the model.*

Thank you for your suggestions. The reviewer's main point is that one can use adaptive or much smaller time steps with explicit schemes so that implicit schemes no longer have an advantage. Well, yes and No. We have several points of response:

(1) Yes, you can reduce time steps, but with automatic differentiation (AD), each step (especially those with thresholds) incur memory usage, CPU overhead and add to the length of the gradient chain, in addition to adding to the computational expenses during forward. Many times we need threshold functions even with a small time step because some operations like logarithm cannot admit zero or the smallest negative values. Using a small time step will incur more memory use.

(2) We agree that explicit schemes are valuable can be used in many cases, but it has been studied extensively in numerical algorithms that stiff ODEs are best handled by implicit schemes (Sundnes, 2023, https://en.wikipedia.org/wiki/Stiff_equation). These numerics are

documented in many decades of literature and we believe it no longer requires us to prove it. Here we are saying we must enable implicit solvers, not saying explicit solvers cannot be used. They both have their advantages and disadvantages.

(3) Batch-dimension parallelism is absolutely crucial because the point of differentiable modeling is to support big-data learning. Batch enables learning across many basins or instances, but the solver may also run into different numerical characteristics and time-stepping requirements, rather than the uniform operations preferred by the GPU. This is why adaptive time stepping is tricky for running differentiable models with minibatch and the adjoint solves an important problem. In fact, we have tried adaptive explicit ODE solvers, and while they work beautifully for one problem, they do not work well for parallel simulations with a batch. Furthermore, to use small time steps rigorously, in theory you need to match forcing inputs to those tiny steps, which requires interpolation schemes and potentially adds lots of complexity.

We will add these points into the following paragraph to the revised paper:

"*While this paper focuses on enabling implicit solvers in differentiable modeling, we do not suggest that explicit solvers are to be discouraged. Runge-Kutta schemes can be well suited for a number of cases and may be attempted for the rainfall-runoff case. It has long been explored in the numerical algorithm literature that each type of solvers has their advantages and disadvantages and is suitable for different problems. For example, implicit solvers are not only preferred but also necessary for stiff ODEs, especially those with dynamics on vastly different time scales and those resulting from the discretization of elliptic PDEs. Using explicit solvers for them could necessitate very small time steps which need to be coordinated with the modification of forcing inputs. In the context of differentiable modeling, a new dimension of consideration plays an important role --- GPU parallel efficiency at the batch level --- because the primary point of differentiable modeling is to learn from big data. Either explicit or implicit scheme needs to serve this purpose. This means that time-adaptive solvers that may require vastly different time steps amongst batch members may have limited applicability when we want to use minibatches. In addition, as discussed in the Introduction, all automatic differentiation steps incurs CPU overhead and storage burdens --- thresholds and array mutation, especially, often require data storage on the GPU. GPU memory may soon run out if we have too many iterations, either with explicit or implicit schemes, which could limit the training lengths. If neural network weights participate in the calculations of these iterations, it further induces the problem of vanishing gradients. We need to put these constraints into consideration and design balanced algorithms.*"

We appreciate the suggestion, given our above suggested revision that *we are here to **enable** implicit solvers but this should not discourage explicit solvers*, we believe it is unnecessary for us to run the model at extremely small time steps to prove the point. In fact, we continue to use our sequential code with the understanding that it gives us a bit higher efficiency but a bit lower numerical performance. If we run many many small time steps, we gain back numerical performance but then lose back efficiency.

In fact, as explained immediately above, running the model at extremely small time steps with AD, especially when you have operations that require data storage (you need this nonetheless as you cannot allow negative values in some operations and explicit algorithms cannot guarantee nonnegativeness), is impractical for explicit solvers due to GPU memory

usage, for the same reason many iterations pose problems for implicit schemes. It has also been shown before differentiable modeling that fixed-step explicit schemes with shorter time steps only provide a poor balance between accuracy and efficiency (Clark et al. 2010).

Running this model on a fine time step also seems not to make much sense ---- if you run on hourly or minute time scale, you are supposed to also need at least hourly inputs which requires more data preprocessing work, and will most likely run out of GPU memory before running the model for a year. It also seems that it should not be our group's responsibility to provide something like a parallel adaptive explicit solver --- in fact this could be quite hard to do: running time-adaptive solvers may also run into challenges to batch-level GPU parallel efficiency for the purpose of learning from big data. Some adaptive schemes that work well for individual basins may not work for the batch on the GPU. Running on CPU in MPI could work, but it is substantially more involved in coding and comes at 2-3 orders of sacrifice in energy cost, which most people in machine learning do not want to do.

Given these considerations, we did not implement adaptive time stepping methods initially (we actually tried this with another package in Julia but such algorithms did not support high GPU parallelism across the batch members, so we settled for pytorch and implementing our own solvers). But if not adaptive, how small a time step is enough? Tiny time steps kill the GPU ram. We shouldn't be micromanaging the time step for each different case we run. The PI here admits that the choice of these solvers and algorithms in fact resulted from quite some elaborate exploration and messing around with various alternative Scientific Machine (SciML) packages and since 2021 during his sabbatical time, and there are many reasons why we settled on our choices.

If the editor insists that we try small time steps, we could give it an earnest attempt, but we think **it would be a little bit unfair to put this responsibility on us**, while delaying us from working on other important problems we think that need to be addressed in this new domain. We very much welcome the community to contribute to the comparisons, as there is enormous space here for the next developments. Hence, while we very much appreciate the constructive opinions, we respectfully disagree with the reject recommendation. We suggest that the above two issues raised by Dr. Ehret could be addressed by revising the manuscript, making clarifications and stating limitations, as the paragraph proposed above.

Unfortunately over the AGU and winter break time frame the interactive discussion has ended, we wonder if we could discuss more about this.

Reference:

Clark, Martyn P., and Dmitri Kavetski. "Ancient numerical daemons of conceptual hydrological modeling: 1. Fidelity and efficiency of time stepping schemes." *Water Resources Research* 46, no. 10 (2010).

Kavetski, D. and Clark, M.P., 2010. Ancient numerical daemons of conceptual hydrological modeling: 2. Impact of time stepping schemes on model analysis and prediction. *Water Resources Research*, *46*(10).

Sundnes, J., 2023. Solving Ordinary Differential Equations in Python (Vol. 15). Springer Nature.

Here are our response to the detailed comments in part A from Dr. Ehret:

6

*"Therefore, in this manuscript, in addition to the description of how to include implicit schemes in MLworkflows, I was expecting a demonstration of how this actually solves a problem."*

In Section 2.2.4, 'Backpropagation with a Coupled Neural Network and Process-Based Model', and Section 2.2.5, 'Adjoint-Based Implicit Scheme', we demonstrate step-by-step how implicit schemes are derived and function within differentiable models. This functionality is primarily numerical and can only be evidenced through changes in model performance, in contrast to model structures that have physical meanings.

*"The authors conclude in their study that the (small) model improvements between the HVB-hybrid variants using explicit and implicit schemes are due to problems introduced by the explicit scheme (lines 412-415), but because they do not provide a proof for a cause, the conclusion based on an effect is not convincing."*

In this study, we conducted a rigorous comparison between implicit and explicit schemes. The structure of the HBV model, the hyperparameters used in the embedded neural network, and the datasets remained consistent across both approaches. The only variable was the numerical approximation. If there is an improvement in model performance, we believe it can be attributed to the reduction of numerical errors.

*"The authors mention that computational costs for the latter increased by a factor of 5-10 (line 581). Increasing the time stepping of the explicit scheme from daily to 6 hours would only mean a factor of 4, but would already resolve diurnal cycles, which might be relevant additional information for the model."*

The computational costs are not solely due to the iteration steps in Newton's iteration but also because of the calculation of the Jacobian matrix in backpropagation. The Newton-Raphson solver can converge within an average of 3-4 iterations. Its forward computational cost is comparable to a 6-hour time-stepping scheme. However, the issue extends beyond computation cost and memory usage for storing gradients of each operation. More critically, it involves the potential for gradient vanishing or explosion – a well-known problem in machine learning – due to the accumulation of gradients with AD over all time steps and iterations in a training instance. The adjoint method for the implicit iterative solver can bypass gradient tracking in the iterations of each time step, which helps mitigate this problem.

*B) Motivated by problems of the HBV model to simulate (near-)zero base flow during extended dry spells, the authors integrate a detailed study about the effect of adding an additional capillary rise process to the HBV model. This is a valid question and analysis, but it does not at all support the main argument of the paper about how and why implicit schemes can be integrated into modern hybrid modeling workflows. I therefore suggest presenting this analysis in another paper, and removing it from this one. In this context, it is interesting that the authors provide a range of possible adjustments to the HBV model to help it achieve (near-)zero flow (strategies 1-5 in lines 258-261; and lines 479-482). These adjustments touch very different physical subdomains and processes of the model, and one may wonder about the limitations of a supposed key advantage of physics-based models – realism and interpretability – if it remains mainly up to the user's preference which one is*

*chosen. In particular, I wonder why capillary rise from the lower subsurface should bypass the upper surface and directly connect to the surface soil moisture storage, and why the authors chose it this way. Based on the above points, my overall recommendation is that the key topic of the paper is worth publication, but also that the required changes will require time. Therefore I recommend rejecting the paper in present form, but strongly encourage a resubmission.*

*Yours sincerely,*

*Uwe Ehret*

(B) Again, while we respect the opinion of Dr. Ehret, we doubt if deleting the change is the correct course of action. Many article carry more than one stories and this is a beneficial (although not that major) improvements to the model. We do not want to write another article for this change. Also, the differences between these models are very well documented ---- that is, the performance differences only from sequential to adjoint, and then from adjoint to the adjoint model with improvement structure are clearly provided in clear detail. The readers need to know what is the impact of solution accuracy, and this second story put things into context regarding how that impact compares to a change in the structure. After fixing the parameter and numerical errors, the model is now ready to understand the defects in its structure. Previously, numerical errors, parameter errors, and model structure errors were intertwined. Now, we are able to separate them and learn new physics and compare their effects. Hence, we think this is quite a useful comparison and should be retained.

We used the additional capillary rise process as an example to show how the differentiable model can be further improved with structural modifications. The current structure of capillary rise is learned from the GSFB model, which represents the recharge from deep groundwater. We think it is more appropriate to term it 'capillary rise' (Ye et al. 1997, Model 20 in Knoben et al. 2019). The fact that this component connects back to the surface reduces the complexity and reflects that some shallow subsurface flow (like lateral soil interflow) can indeed bypass the upper subsurface by following preferential flow paths. These are conceptual models that cannot fully take into account the spatial heterogeneity so some effective representation is needed.

Ye, W., Bates, B.C., Viney, N.R., Sivapalan, M. and Jakeman, A.J., 1997. Performance of conceptual rainfall-runoff models in low-yielding ephemeral catchments. *Water Resources Research*, *33*(1), pp.153-166.

Knoben, W.J., Freer, J.E., Fowler, K.J., Peel, M.C. and Woods, R.A., 2019. Modular Assessment of Rainfall–Runoff Models Toolbox (MARRMoT) v1. 2: An open-source, extendable framework providing implementations of 46 conceptual hydrologic models as continuous state-space formulations. *Geoscientific Model Development*, *12*(6), pp.2463-2480.

---

## Author Comment (AC4)

We appreciate the comments from the reviewer. We still want to keep our paper as a research paper. We plan to add a more in-depth discussion to show the importance of the implicit scheme in large-scale hydrological simulation with big data and how our results align with the findings of Clark and Kavetski (2010) and Kavetski and Clark (2010), but we do not want to repeat their work that has been done thoroughly.

Since daily forcing and streamflow data are more commonly available and accessible, while hourly input data is often harder to obtain, in this work, our focus is solely on daily hydrological modeling using conceptual models. The numerical schemes employed within the framework of the differentiable model need to maintain stability for simultaneous large-scale simulations in minibatch while also allowing for gradient tracking. We conducted tests on the differentiable HBV model with various numerical schemes and time steps. In theory, with smaller time steps, we were supposed to configure forcing functions to provide hourly inputs that match the progression of time within a day, i.e., the inputs should reflect diurnal changes in forcing. However, as the datasets do not contain such hourly information and we are only doing a numerical comparison, we chose not to do that here.

We plan to include the following analyses in our discussion section (potentially as an appendix):
1. The fixed-step explicit Euler scheme with one day time step caused divergence in the large-scale simulation due to its instability. However, with a 4-hour time step, it exhibited better performance than the 4th-order Runge-Kutta (RK) explicit scheme but still lagged behind the sequential scheme that employed ad-hoc operation splitting (attached Table 1). The ad-hoc operation splitting updates the state variables after each operation has higher accuracy and stability. The implicit scheme yielded superior performance in terms of KGE and high and low flow metrics. Reducing time steps and using implicit schemes both improved model accuracy, aligning with the findings of Clark et al. in 2010.

2. An important finding in Kavetski and Clark (2010) is that numerical approximation errors can be compensated for by distorted parameter values during calibration, resulting in a 'right result for the wrong reasons.' This phenomenon can affect parameter uncertainty analysis, hinder meaningful parameter interpretation and regionalization, and lead to erroneous internal model dynamics. To explore this further, we examine the metric (KGE) surface within the 2D slice defined by field capacity ($\theta_{FC}$) and the parameter $\beta$, derived from various numerical solutions (attached Figure 1). Thanks to the neural network's ability for parameter regionalization, we did not observe a notable macroscale roughness as shown in Figure 1 of Kavetski and Clark (2010) when using explicit schemes. However, we still observed distortions and roughness in the KGE surface in models employing the RK scheme (site A and D). As we reduced the time steps and transitioned to implicit schemes, these distortions seem to have alleviated and converged toward the metric surface consistent with the correct numerical solution. That is, the 4-hourly patterns are more similar to the implicit results than the daily, and we expect further convergence when we are able to run the one-hourly model (this takes more time to prepare). The convergence toward the implicit scheme suggests that the implicit scheme results were more reliable.

3. Even with small time steps, we still observe that explicit schemes exhibit parameter distortion and potentially problematic internal model dynamics, as demonstrated in Table 2 and Figure 6 in the manuscript. Such issues have implications on our ability to learn a better model structure. Hence, we argue that implicit scheme has value.

4. Regarding the reviewer's recommendation to drop the analysis about the structural change, we have some reservations. Because we are providing comprehensive comparisons both with the change and without the change, it does not seem like the readers would lose anything by seeing the best configurations. In fact, it is to the benefit of the community to see what changes matter and by how much, and how to obtain the state of the art. We promise to present it in a clear and insightful way. While we fully respect the reviewer's opinion and could see where he is coming from, we respectfully insist on our paper's design here.

Table 1: Summary of streamflow metrics for models using different numerical schemes. As 1-hour time-step models take much more time, the comparison may be provided later in the revision, potentially as an appendix. Upon revision, a comparison of timing and memory use will also be provided.

| Model | Numerical scheme | Time step | Median NSE | Median KGE | Median low flow RMSE (mm/day) | Median peak flow RMSE (mm/day) |
|---|---|---|---|---|---|---|
| δHBV | Fixed-step explicit | 1 day | - | - | - | - |
| δHBV | The fourth-order Runge-Kutta explicit | 1 day | 0.69 | 0.70 | 0.061 | 3.25 |
| δHBV | Fixed-step explicit | 4 hours | 0.72 | 0.71 | 0.09 | 2.50 |
| δHBV | Sequential | 1 day | 0.73 | 0.73 | 0.074 | 2.56 |
| δHBV.adj | Implicit adjoint | 1 day | 0.72 | 0.75 | 0.048 | 2.47 |

[Figure]

**(I) 4th order Runge-Kutta explicit scheme**

[Figure]

**(II) Fixed-step Euler explicit with 4 hour time step**

**(III) Sequential scheme**

**(IV) Implicit adjoint scheme**

**Figure 1: Impact of numerical schemes on the KGE surface of HBV model: The contour of KGE calculated from the (I) 4th order Runge-Kutta explicit scheme, (II) Fixed-step Euler explicit with 4 hour time step with 4 hour time step, (III) sequential scheme, and (IV) implicit adjoint scheme on the 2D slice of field capacity ($\theta_{FC}$) and parameter β. The predicted parameter values are positioned at the central point of the contours delineated by circles. The locations of selected sites are annotated in Figure 6**

---

## Author Response (AR1)

*Editor*⌗

*Dear Song et al.,*
*First of all, I would like to commend the authors for this interesting work, which aligns well with the scope of HESS. Furthermore, I would like to underscore the great discussion with both reviewers. In your last response to Dr. Ehret, you outlined a promising strategy for enhancing your manuscript, and I eagerly anticipate a revised and more streamlined version.*

*Two minor comments to augment the reviewers' feedback:*

*1. The term "outperform" and the pronounced emphasis on model comparison appear somewhat misaligned with your model's results and also seem to occupy excessive space in this manuscript. How much does your model's performance vary by merely re-training with an alternative weight initialization method, adjusting your hyperparameters, or by interchanging the training and testing datasets? My intention is constructive; I am not suggesting further tests but rather expressing that the approach and concept itself are intriguing and novel. I would have appreciated a deeper explanation and discussion on how this work could be extended to problems beyond bucket models or / and a nice schematic figure with some extra text how this is implemented in the network. Particular because your code is not shared yet which seems unnecessary because the preprint in HESS is open.*

*2. "Since daily forcing and streamflow data are readily available and accessible, while hourly input data is often more challenging to acquire, our study focuses exclusively on daily hydrological modeling using conceptual models." A quick Google search (10.5281/zenodo.4072700) revealed the existence of datasets (as well as a API of the USGS), and it's likely there are more I am unaware of. This is merely a suggestion should you wish to conduct your experiments at an hourly resolution. I do not think these experiments are necessary here (although interesting); however, I found the argument regarding the lack of data to be somewhat tenuous.*

*Again congratulations on the nice work and I look forward to the revised manuscript.*

*Sincerely,*

*Ralf Loritz*

Dear editor,

Thank you for your comments. The effects of the weight initialization method (random number selection) or adjusting hyperparameters are negligible, as this work has been done by Feng et al., 2022, regarding the sequential model. Our adjoint model inherited their hyperparameter configuration to ensure a fair comparison and we did not attempt hyperparameter tuning. Limited tuning suggests the results to be stable. We have added Figure 2 to provide a schematic representation of how the adjoint is utilized within the code for better understanding. The code for the adjoint differentiable models has updated as an attachment for reviewers.

We have revised the manuscript based on the responses proposed in our discussion with the two reviewers. We appreciate their valuable feedback, which prompted us to further

consider the coordination between the numerical schemes, the temporal resolution of forcing, and the parameter learning functions (neural networks) in the differentiable hydrological models. We conducted tests on the differentiable HBV model with various numerical schemes and fixed smaller time steps (Table A2 & Figure A3 in Appendix)  We added the following main points in our discussion section:

- Directly training an hourly model with ML techniques remains computationally expensive and may lead to the issue of gradient vanishing if the time step is too small, according to the discussion in Gauch et al. (2021). In the literature, some ML techniques have been used to predict hourly flood hydrographs using daily flow data, which require further investigation in the differentiable models.
- Our new results added to the Appendix shows the sequential model and implicit adjoint model with a 1-day time step have higher performance than the explicit Euler schemes with smaller time steps or the fourth-order Runge-Kutta scheme when using daily inputs.
- The forcing and physical parameter configuration in the differentiable model might need to match the timesteps of the numerical schemes to reflect the hydrograph changes within a day if smaller timesteps are used.
- The adjoint used in this work is derived for the gradient of the Newton solver, such that it can theoretically support any model that can be solved with the Newton solver—not only bucket models governed by ODEs but also distributed models governed by PDEs. However, the challenge may still exist in calculating the Jacobian matrix for a batch of basins for PDEs, as the distributed parameters in PDEs are significantly greater in number than those in ODEs, and can slow down the efficiency of the model in both forward and backward modes.

Thank you and the reviewers for your constructive feedback!

Chaopeng

*Reviewer #1*

*Dear authors, Dear editor,*

*Here is my review of the submitted work. I recommend accepting the manuscript after minor revisions.*

*Kindly*

*Ilhan Özgen-Xian*

*General comments and questions*

*1. The authors convincingly make an argument for implicit time integration.  The forward Euler time stepping used in this work is indeed at a disadvantage if fixed time steps are used.  However, it is not clear to me how higher order explicit time integration methods such as schemes from the explicit Runge-Kutta family (RK) would perform in comparison to the implicit one.  If I understood correctly, some of the numerical issues mentioned in the manuscript might also be addressed by (adaptive) multistep schemes of this type.  The*

*advantage of RK-type schemes in this context is that the number of computations per time step is known a priori. In contrast, the Newton-Raphson iterative solver may require any number of steps until convergence. High order RK schemes, for example the standard RK45 or the adaptive RK-Fehlberg method, could also potentially benefit from the adjoint method presented in this paper to avoid excessive memory usage. Perhaps the authors can comment on this.*
*2. The authors mention that the Newton-Raphson solver introduces some overhead to the computation. On average, in the results shown in this paper, how many iteration steps were necessary for the solver to converge?*

Thank you for your suggestions. We believe these two questions can be addressed together. The Newton-Raphson solver typically converges in 3-4 iterations on average (information added in line 611). The number of iterations or steps in the Newton solver and the RK scheme are comparable. The primary computational burden of the implicit scheme lies in calculating the Jacobian matrix for the Newton solver, whereas explicit schemes only require forwarding through the physical model. We tested both the 4th order RK scheme and the explicit Euler schemes with fixed, smaller time steps (4 hours and 1 hour). Their results are reasonable but not as robust as the sequential model and implicit adjoint model with one day time step we reported in the main text (please refer to Table A2 and Figure A3 in the Appendix which shows that their metrics are not better than the sequential and implicit models). The most likely reason is that the daily forcing inputs and physical parameters from the neural network do not align with the smaller time steps within a day. The neural network for parameterization is configured to match the temporal resolution of the input data, which is one day. Consequently, the forcing and physical parameters remain constant within a day, failing to capture diurnal changes in forcing. Using explicit solvers with smaller time steps within the differentiable model framework needs to be coordinated with modifications to the forcing inputs and training target data. However, even though hourly data are now publicly available, directly training an hourly model using ML techniques are more computationally expensive and may lead to the well-known issue of gradient vanishing (Gauch et al., 2021). Some ML techniques, such as multi-time-scale learning, have been considered for converting daily flow data into hourly flood hydrographs (Gauch et al., 2021; Sarıgöl and Katipoğlu, 2023). We have an ongoing study working on this topic.

We will add some explanations in our discussion: *"While this work focuses on enabling implicit solvers in differentiable modeling, we do not suggest that explicit solvers are to be discouraged. It has long been explored in the numerical algorithm literature that each type of solver has advantages and disadvantages and is suitable for different problems. For example, implicit solvers are not only preferred but also necessary for stiff ODEs, especially those with dynamics on vastly different time scales and those resulting from the discretization of elliptic PDEs. Using explicit solvers for them could necessitate very small time steps.*

*Further complications of using explicit schemes with small time steps include computational expenses, parallel efficiency, and matching forcing functions. Even though hourly data are now publicly available, directly training an hourly model with ML techniques remains computationally expensive and may also cause the notorious problem of gradient vanishing if the training time steps are too numerous (Gauch et al., 2021; Greff et al., 2017). The numerical schemes employed in the physical models within the differentiable modeling framework need to maintain stability for simultaneous large-scale simulations in each minibatch while also allowing for gradient tracking.*

*Batched learning and parallel efficiency may prefer uniform operations across basins and challenge the application of adaptive time-stepping algorithms. We conducted tests on the differentiable HBV model with various numerical schemes and fixed smaller time steps (Table A2 & Figure A3 in Appendix). The sequential model and implicit adjoint model with a 1-day time step presented higher performance than the explicit Euler schemes with smaller time steps or the fourth-order Runge-Kutta scheme. The main reason may be that the daily forcing inputs and daily physical parameters from the neural network do not match the smaller time steps within a day. Thus, explicit schemes with smaller time steps may require matching forcing functions as well. Some multi-time-scale ML techniques have been used to predict hourly flood hydrographs using daily flow data to avoid gradient vanishing issues in the direct hourly training (Gauch et al., 2021; Sarıgöl and Katipoğlu, 2023). These approaches present possible solutions for future investigations."* Line {570-590}

Gauch, M., Kratzert, F., Klotz, D., Nearing, G., Lin, J., Hochreiter, S., 2021. Rainfall–runoff prediction at multiple timescales with a single Long Short-Term Memory network. Hydrol. Earth Syst. Sci. 25, 2045–2062. https://doi.org/10.5194/hess-25-2045-2021

Sarıgöl, M., Katipoğlu, O.M., 2023. Estimation of hourly flood hydrograph from daily flows using machine learning techniques in the Büyük Menderes River. Nat. Hazards 119, 1461–1477. https://doi.org/10.1007/s11069-023-06156-x

*Minor comments*

*1. P.2, L.70: "graphical processing units" should be "graphics processing units"*

Revised as suggested.

*2. P.3, LL.105ff.: Does "elliptic operator" in this context correspond to the Laplacian? If so, some of the examples might require some annotation. The Saint-Venant equation only contains Laplacian operators if molecular/turbulent diffusion is accounted for. Many forms of the Saint-Venant equation omit these terms, for example (García-Navarro et al., 2019, doi:10.1007/s10652-018-09657-7; LeVeque et al., 2011, doi:10.1017/S0962492911000043).*

We revised it to "shallow water equations", which refers to a two-dimensional Saint-Venant equation considering turbulent diffusion.

*3. P.3, LL.105ff. (continued) When I looked at the paper by Aboelyazeed et al. (2023) (cited by the authors), I couldn't see Laplacians in the Farquhar model equations.*

Aboelyazeed et al. (2023) is an example of systems of nonlinear equations, not an elliptic operator.

*4. P6, L.209: "The same forcings ... was used" should be "The same forcings ... were used"*

Revised as suggested.

*5. P.12, L.335: Should it be Eq. (28) instead of Eq. (27)? May be I am misunderstanding something.*

Yes, your understanding is correct. We fixed it.

*6. P.14, L.398: The authors state that the mass balance preservation of the adjoint-driven NN-HBV model might be the reason behind the improved model performance. I don't understand why the mass conservation should significantly differ from the explicit sequential NN-HBV model if the hydrological process representation remains untouched. Is this related to the use of thresholds to avoid negative storages? Can the authors elaborate a bit more?*

Yes, by avoiding thresholds for negative states, the implicit model can achieve better mass conservation. This impact is significant for low flows. For example, the thresholds for lower subsurface zone storage in the current HBV model can induce minimal baseflow on dry days. More importantly, the adjoint (implicit) model greatly reduces numerical errors, thus improving the model's performance. We revised the main text to: *"The advance may be attributable to HBV.adj's reduction of numerical errors, which forces the model to more accurately represent extreme values."*

*7. P.24, L.580: The additional computational cost introduced by the implicit solver is quite substantial (18 h vs. 133 h), suggesting either poor convergence or large communication overhead in the implicit scheme.*

We agree with the reviewer that the computational cost of the implicit solver is substantial compared to the sequential model. However, when compared with traditional models that require basin-by-basin calibration on a CPU, it is efficient for large-scale modeling. The implicit solver can converge in 3-4 iterations, and all training is conducted on a single GPU, with no need for communication between nodes. The reasons it is slower than the sequential model are twofold: 1) the HBV model is called 3-4 times in each time step, whereas the sequential model only needs to call it once, and 2) the calculation of the Jacobian matrix for multiple basins, depending on the batch size, also consumes time. There are additionally some CPU overhead issues to be explored down the road.

**Reviewer #2**

*Dear Chaopeng Shen, dear Authors, dear Editor,*

*Thanks for your detailed replies to my review. This is exactly what the HESS open discussion forum is made for. First of all, I would like to state once more that I think the authors, by introducing in their manuscript a method for integrating implicit solvers in modern ML-based model training via backpropagation, provide valuable research that is absolutely worth publishing. The reason I was (and am) recommending "reject with strong encouragement for resubmission" is that I suggest changes to the manuscript that will most likely require more time than usually assigned for major revisions. I will leave it to the Editor whether he thinks i) the changes I suggest are valid, and if yes, ii) how much time they would require. Reading the replies (AC2, text and .pdf, and AC3) by Chaopeng Shen, four main points of discussion arise. I reply to them here in summarized form, rather than individually in each document:*

- ***The main purpose of the paper is to enable implicit schemes.*** *I agree with this statement by Chaopeng Shen, and I suggest focusing on this message. Therefore, I would support a revised paper, as a technical note, that introduces the method, not more and not less. In such a paper, there is no need for an in-depth discussion about if and when explicit schemes for conceptual hydrological models operating on daily data create problems, and no need for comparing implicit schemes vs. explicit schemes operated on higher-resolution data comparison.*
- ***Running small time steps (less than a day) with automatic differentiation creates problems (memory use, allowable window size).*** *Chaopeng Shen suggests it is a bit unfair by me to ask the authors to demonstrate that implicit schemes solve a problem of explicit schemes. I see two options here. The first is to write a short technical note presenting only the main method innovation, see previous bullet point. The second is to keep it as a research paper, showing the method innovation and applications. In that case, as a reader I would expect a demonstration that the innovation solves an important problem present in the applications used in the paper (hydrological modeling on daily basis using conceptual models). That is, showing that the currently used explicit schemes introduce substantial numerical error. This does not need to be for the full set of catchments used, but could be done for a few representative catchments, and along the lines of the demonstrations in Clark and Kavetski (2010). If the editor thinks this is unnecessary detail, I would at least expect a more in-depth discussion about how the findings of Clark and Kavetski (2010) and Kavetski and Clark (2010) apply to the application in the manuscript.*
- ***Adaptive time stepping is difficult to realize in connection with AD, more specific in connection with parallel processing of minibatch optimization.*** *I never tested, but Chaopeng Chens explanation of this point makes sense to me. In my review, I never asked the authors to include such tests in the manuscript, therefore there is no disagreement here.*
- ***Discussion of HBV structural/functional changes (capillary rise) in the same paper.*** *In my review, I was mentioning that discussing structural/functional changes to the HBV model to solve an apparent model deficiency has little to do with the key message of the manuscript, and therefore suggested removing it. I still think that leaving this part away will help the paper to better convey its message. The reply by Chaopeng Shen - "Many article carry more than one stories and this is a beneficial (although not that major) improvements to the model. We do not want to write another article for this change." - has not convinced me otherwise. I will leave this decision to the editor.*

*Yours sincerely, Uwe Ehret*

We appreciate the comments from the reviewer. We still want to keep our paper as a research paper. We plan to add a more in-depth discussion to show the importance of the implicit scheme in large-scale hydrological simulation with big data and how our results align with the findings of Clark and Kavetski (2010) and Kavetski and Clark (2010), but we do not want to repeat their work that has been done thoroughly.

In this work, our focus is solely on daily hydrological modeling using conceptual models. The numerical schemes employed within the framework of the differentiable model need to maintain stability for simultaneous large-scale simulations in the minibatch while also allow for gradient tracking. We conducted additional tests on the differentiable HBV model with various numerical schemes and time steps (see appendix Table A2 and Figure A3). In theory, with smaller time steps, we were supposed to configure forcing functions to provide hourly inputs that match the progression of time within a day, i.e., the inputs should reflect diurnal changes in forcings. Even though hourly data are now publicly available, directly training an hourly model with ML techniques remains computationally expensive and may also cause the notorious problem of gradient vanishing (Gauch et al., 2021).

We included the following analyses in our discussion section and appendix:
1. *"We conducted tests on the differentiable HBV model with various numerical schemes and fixed smaller time steps (Table A2 & Figure A3 in Appendix). The sequential model and implicit adjoint model with a 1-day time step presented higher performance than the explicit Euler schemes with smaller time steps or the fourth-order Runge-Kutta scheme. The main reason may be that the daily forcing inputs and daily physical parameters from the neural network do not match the smaller time steps within a day. Thus, explicit schemes with smaller time steps may require matching forcing functions as well. Some multi-time-scale ML techniques have been used to predict hourly flood hydrographs using daily flow data to avoid gradient vanishing issues in the direct hourly training (Gauch et al., 2021; Sarıgöl and Katipoğlu, 2023). These approaches present possible solutions for future investigations." line{583-590}.*
The implicit scheme yielded superior performance in terms of KGE and high and low flow metrics. Both reducing time steps and using implicit schemes improved model accuracy, aligning with the findings of Clark et al. in 2010.

2. An important finding in Kavetski and Clark (2010) is that numerical approximation errors can be compensated for by distorted parameter values during calibration, resulting in a 'right result for the wrong reasons.' This phenomenon can affect parameter uncertainty analysis, hinder meaningful parameter interpretation and regionalization, and lead to erroneous internal model dynamics. To explore this further, we examine the metric (KGE) surface within the 2D slice defined by field capacity ($FC$) and the parameter, derived from various numerical solutions (Figure A3). *"The parameterization function (the neural network) embedded in the differentiable models demonstrates robustness, as evidenced by the similarity of parameter patterns and metric surfaces derived from various numerical schemes in Figure 7 and Figure A3. We did not observe a notable macroscale roughness in the metric surface (Figure A3) as shown in Kavetski and Clark (2010) when using explicit schemes. Moderate distortions and roughness were present on the KGE surface in models employing the RK scheme (sites A and D). As we reduced the time steps and transitioned to implicit schemes, these distortions seem to have alleviated and converged toward the metric surface, consistent with the correct numerical solution. That is, the 4-hourly and hourly patterns are more similar to the implicit results than that of the RK scheme. The convergence toward the implicit scheme suggests that the implicit scheme results are more reliable." line{591-599}*

3. Even with small time steps, we still observe that explicit schemes exhibit parameter distortion and potentially problematic internal model dynamics, as demonstrated in Table 2 and Figure 7 in the manuscript. Such issues have implications on our ability to learn a better model structure, and hence we argue that the implicit scheme has value.

4. Regarding the reviewer's recommendation to drop the analysis about the structural change, we have some reservations. Because we are providing comprehensive comparisons both with the change and without the change, it does not seem like the readers would lose anything by

seeing the best configurations. In fact, it is to the benefit of the community to see what changes matter and by how much, and how to obtain the state of the art. While we fully respect the reviewer's opinion and could see where he is coming from, we respectfully would like to continue including this content. Nevertheless, in response to the reviewer' comments, we reduced the amount of discussion with respect to this component.

**Table A2: Summary of streamflow metrics for models using different numerical schemes and time steps. Timing was obtained on a Nvidia Tesla V100 GPU.**

| $\delta$Model | Numerical scheme | Time step | Memory Usage per batch | Computational time per batch | Median NSE | Median KGE | Median low flow RMSE (mm/day) | Median peak flow RMSE (mm/day) |
|---|---|---|---|---|---|---|---|---|
| $\delta$HBV | Fixed-step explicit | 1 day | 2274M | 1.6s | - | - | - | - |
| $\delta$HBV | The fourth-order Runge-Kutta explicit | 1 day | 2532M | 3.9s | 0.69 | 0.70 | 0.061 | 3.25 |
| $\delta$HBV | Fixed-step explicit | 4 hours | 2706M | 6.3s | 0.72 | 0.71 | 0.09 | 2.50 |
| $\delta$HBV | Fixed-step explicit | 1 hours | 4146M | 18.1s | 0.72 | 0.71 | 0.08 | 2.63 |
| $\delta$HBV | Sequential | 1 day | 2266M | 1.5s | 0.73 | 0.73 | 0.074 | 2.56 |
| $\delta$HBV.adj | Implicit adjoint | 1 day | 2788M | 19.5s | 0.72 | 0.75 | 0.048 | 2.47 |

[Figure]

**Figure A3: Impact of numerical schemes on the KGE surface of the HBV model: The contour of KGE calculated from the (I) 4th order Runge-Kutta explicit scheme, (II) Fixed-step Euler explicit with 4 hour time step with 4 hour time step, (III) Fixed-step Euler explicit with 1 hour time step, (IV) sequential scheme, and (V) implicit adjoint scheme on the 2D slice of field capacity (FC) and parameter. The predicted parameter values are positioned at the central point of the contours delineated by circles. The locations of selected sites are annotated in Figure 7.**